# DOES YOUR 3D ENCODER REALLY WORK?
# A SIMPLE YET EFFECTIVE PATHWAY TO REAL 3D SCENE UNDERSTANDING

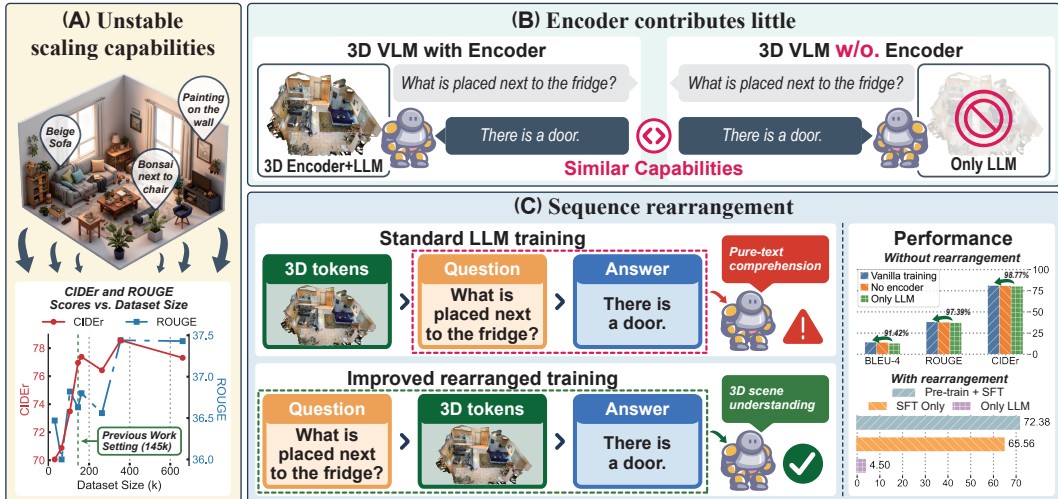

Figure 1: **(A) Unstable scaling capabilities:** gradually increasing the scale of training data demonstrates unstable data scaling capabilities of 3D VLM. **(B) Encoder contributes little:** further experiments reveals that the model's performance is similar with and without the 3D encoder. **(C) Sequence rearrangement:** After determining that this issue stems from model overfitting, we address it by rearranging the input sequence to disrupt the continuous Question-Answer text. 3D VLMs with only LLM can achieve 98.77% performance compared to vanilla training, while only maintain 6.51% pefromance after sequence rearrangement.

## ABSTRACT

Remarkable progress in 2D Vision-Language Models (VLMs) has spurred interest in extending them to 3D settings. Based on their encoder design, this paper categorizes recent 3D VLMs into 3D scene-centric, 3D object-centric and 2D image-based approaches. Despite their architectural similarity to 2D counterparts, 3D scene-centric VLMs have exhibited comparatively lower performance compared with the latest 3D object-centric and 2D image-based approaches. To understand this gap, we conduct an in-depth analysis and find that these models show unstable data scaling capabilities and limited reliance on the 3D scene encoder, instead overfitting to linguistic cues and frequent answers. Although data balancing of under-sampling offers partial improvements, it fails to address the fundamental problem, as the model continues to largely ignore the 3D scene input. To address these limitations and encourage genuine 3D scene understanding, we introduce a simple yet effective training strategy: **rearranging the input sequence**. By positioning the 3D scene between the question and the answer, we prevent the model from learning shortcuts from linguistic cues alone and compel it to ground its comprehension in the visual context. Our experiments show this method not only improves the model's genuine understanding, but also restores the effectiveness of standard pre-training and supervised fine-tuning stages. Crucially, our approach ensures the 3D encoder plays an essential role, laying a more robust foundation for future 3D VLM research.

Figure 2: **Visualization of different 3D VLM patterns.** Similar to 2D VLM, 3D VLM also requires an encoder to extract features that serve as 3D tokens for the cross-modal input. Variations in the model design primarily stem from the choice of encoder: (a) directly processes the 3D scene, (b) utilizing a 3D object encoder necessitates initial object detection and subsequent relation modeling and (c) employing a 2D image encoder requires rendering the 3D scene into a sequence of images.

# 1 INTRODUCTION

The remarkable progress of 2D Vision-Language Models (VLMs) through pre-train and supervised fine-tuning (SFT) (Li et al., 2023; Mishra et al., 2024; Chen et al., 2024d; Dong et al., 2024; Liu et al., 2023a; Zhu et al., 2023a; Wu et al., 2024; Wang et al., 2024; Team et al., 2023; Liu et al., 2024) has sparked increasing interest in extending these models to 3D settings (Hong et al., 2023; Wang et al., 2023; Huang et al., 2023a; Li et al., 2024; Huang et al., 2023b; Zhu et al., 2024; Chen et al., 2024a;c; Zhi et al., 2024). By leveraging powerful open-source Large Language Models (LLMs) and richly annotated 3D datasets (Dai et al., 2017; Azuma et al., 2022; Chen et al., 2020; Achlioptas et al., 2020; Ma et al., 2022; Chen et al., 2021), substantial progress has been made in 3D Vision-Language tasks such as **3D Question Answer (3D-QA)** (Azuma et al., 2022; Ma et al., 2022), **3D Dense Captioning (3D-DC)** (Chen et al., 2024b; 2021) and **3D Visual Grounding (3D-VG)** (Chen et al., 2020; Achlioptas et al., 2020).

Unlike the common practice in 2D VLMs of typically utilizing image encoders, 3D scenes, as complex spatial structures comprising various object relationships, can be approached as combinations of different modalities, leading to diverse model design patterns. As shown in Fig. 2, based on the encoder employed, recent works can be categorized into three main types: a) **3D scene-centric VLM** (Hong et al., 2023; Chen et al., 2024a;c; Zhi et al., 2024), which treat each scene as a holistic entity and directly reason about the scene itself, b) **3D object-centric VLM** (Huang et al., 2023b; Wang et al., 2023; Huang et al., 2023a; Li et al., 2024), which understand space as a collection of objects and model individual objects and their relationships; and c) **2D image-based VLM** (Zhu et al., 2024; Zheng et al., 2025b), which interpret space as a continuous images sequence and derive spatial understanding from video analysis.

While 3D scene-centric VLMs are architecturally the most direct extension of successful 2D VLMs, they have generally been outperformed by 3D object-centric and 2D image-based methods on several key tasks (Zheng et al., 2025b; Huang et al., 2024). Consequently, this intuitive design has not emerged as the most prevalent or successful approach in the field. We analyze this performance gap by comparing 3D scene-centric approaches with successful experience from 2D VLMs and begin with three key observations:

- **Observation 1:** *3D scene-centric VLMs fail to demonstrate consistent data scaling capabilities, particularly when trained on large-scale datasets*
- **Observation 2:** *In contrast to 2D VLMs, the pre-train stage appears to have a less significant effect on 3D scene-centric VLMs.*
- **Observation 3:** *3D scene-centric VLMs achieve comparable performance even without the 3D scene encoder.*

Our investigation pinpoints the root cause of these phenomena: a critical over-reliance on linguistic cues and dataset biases. We provide extensive analysis showing that models learn to predict answers directly from questions by memorizing frequent patterns, bypassing the 3D encoder. We verify that this issue is prevalent across various benchmarks and methods, suggesting a fundamental flaw in how these models are commonly trained and evaluated.

To address this, we first explore conventional solutions like data balancing of under-sampling. While experiments shows this offers partial improvements, we find it insufficient to resolve the model's

tendency to ignore the 3D scene. Consequently, we introduce a simple yet powerful solution: **rearranging the input sequence**. By altering the input order from [Scene, Question, Answer] to [Question, Scene, Answer], we disrupt the model's ability to learn direct text-only shortcuts and deteriorate 3D scene understanding to Question-Answer textual understanding. This approach compels the model to ground its reasoning in the 3D scene to make a prediction, thereby ensuring the encoder's role is indispensable.

To summarize, our key contributions lie in:

- We are the first to systematically diagnose and provide quantitative evidence for a critical flaw of ignoring the 3D encoder features in 3D scene-centric VLMs, which achieve 98.77% performance of CIDEr on ScanQA even with only LLM.

- We propose a simple yet effective sequence rearrangement strategy that forces genuine 3D scene understanding. This method prevents the model from relying on text-only shortcuts and makes the 3D encoder's role indispensable, resulting in only 6.21% performance achieved by LLM-only models.

- Our work establishes a more robust training paradigm and pave way to real 3D-awareness for 3D VLMs. We show that under our proposed framework, the distinct and crucial benefits of pre-train and SFT are restored, providing a clear pathway toward developing and fairly assessing models with true 3D spatial intelligence.

## 2 RELATED WORK

### 2.1 3D VISION LARGE LANGUAGE MODELS

The rapid advancement of pre-trained LLMs and their demonstrated strong comprehension and reasoning capabilities have significantly promoted the considerable progress of 3D scene understanding. Based on the 3D encoder employed, existing methodologies can be broadly categorized into three distinct groups:

**3D scene-centric 3D VLM.** With the lack of available 3D scene encoders with rich semantic information, 3D scene-centric methods directly use 3D object detection or 3D scene segmentation models as 3D scene encoders to obtain spatial information. 3D-LLM (Hong et al., 2023) introduces a family of LLM-driven 3D generalist models capable of processing a wide range of textual instructions using 3D features reconstructed from multi-view images. LL3DA (Chen et al., 2024a) leverages Vote2Cap-DETR (Chen et al., 2023; 2024b) to extract scene features and object proposals for the object-centric task. Grounded 3D-LLM (Chen et al., 2024c) proposes Contrastive Language-Scene Pre-training to pre-train a 3D point cloud encoder and a cross-modal interactor for multi-task instruction tuning. LSceneLLM (Zhi et al., 2024) focuses on fine-grained understanding and proposes an adaptive self-attention module and dense vision token selector to dynamically sample question-related tokens.

**3D object-centric 3D VLM.** Object-centric approaches view spatial understanding as dealing with a collection of objects, which enables reusing existing 3D object encoders. These methods usually start by finding all the individual objects in a scene using instance segmentation or object detection. Then, they utilize an available 3D object encoder to get semantic information and a relationship module to model how these objects are spatially related. LEO (Huang et al., 2023b) adopts PointNet++ (Qi et al., 2017) to encode 3D object features and Spatial Transformer (Chen et al., 2022) for modeling point cloud embedding of all objects into object-centric 3D token embeddings. Chat-3D and Chat-3D v2 (Wang et al., 2023) leverage off-the-shelf 3D segmentation models (Jiang et al., 2020; Misra et al., 2021; Qi et al., 2019) for instance segmentation, which is later encoded and modeled through 3D object encoder and relation module to extract scene features. 3DMiT (Li et al., 2024) utilize parallel 3D scene and object encoder (Huang et al., 2022; Xue et al., 2024; Zhou et al., 2023) for global scene and local object visual features.

**2D image-based 3D VLM.** Image-based methods, on the other hand, treat spatial understanding as a video taken in a space (Zhu et al., 2024; Zheng et al., 2025b;a; Huang et al., 2025). This means they can easily connect to existing 2D VLM research as a specific type of multi-view images understanding task. With powerful 2D image encoders, these methods often perform better than those using direct 3D input. LLaVA-3D (Zhu et al., 2024) and Video-3D-LLM (Zheng et al., 2025b)

combines monocular depth and camera pose to obtain spatial position embedding for multi-view image tokens for overall scene understanding.

## 2.2 TRAINING STAGES OF 3D VLMS

The pre-train and supervised fine-tuning (SFT) two-stage training has been shown to work well for 2D VLMs. In the pre-train stage, only the projector between the model and the encoder is trained for better alignment. Then, SFT uses higher-quality and efficient data for instruction tuning. Following this idea, (Zhu et al., 2024; Wang et al., 2023; Huang et al., 2023a;b; Hong et al., 2023; Chen et al., 2024c) train the model with pre-train for alignment and SFT. In contrast, LL3DA (Chen et al., 2024a) only trains the Q-Former (Li et al., 2023) for connecting the 3D encoder and LLM, while (Zhi et al., 2024; Li et al., 2024) directly fine-tunes the projector and LLMs. However, upon closer examination of existing 3D scene-centric approaches, we observe notable variations in the training paradigms employed by models in LL3DA (Chen et al., 2024a), LSceneLLM (Zhi et al., 2024), 3D-LLM (Hong et al., 2023), and Grounded 3D-LLM (Chen et al., 2024c). This divergence in training strategies suggests a lack of unified understanding or consensus within the research community regarding the optimal training methodology for scene-centric 3D scene understanding with LLM, which highlights the need for further investigation into effective and consistent training protocols for this promising direction.

## 3 PROBLEM ANALYSIS

In this section, we first examine how 3D VLMs scale with data, finding unstable data scaling capabilities, especially scaling data during pre-train stage. This prompts us to reconsider the necessity of the encoder, since the purpose of pre-train is to better leverage the encoder's information.

### 3.1 EXPERIMENT SETUP

Our work first focuses on the in-depth analysis of 3D scene-centric approaches. **(1) Benchmark and model:** we select ScanQA as our primary benchmark and adopt LL3DA as our baseline model. LL3DA follows the model design of Fig. 2 (a) with a Q-Former as projector. **(2) Training:** we freeze the LLM and only train the projector during pre-train stage, while fine-tune LLM and projector during supervised fine-tuning (SFT). We default to performing one epoch of both pre-train and SFT stage. **(3) Ablation study:** full model setup represents the original model setting reported officially. No encoder setup denotes only remove the encoder of 3D VLMs, while only LLM setup denotes further removing other module (e.g, position embedding, object IDs) and leaves only the LLM.

### 3.2 DOES PRE-TRAIN STAGE MATTER?

| LLM | Pre-train | SFT | ScanQA | | |
| --- | --- | --- | --- | --- | --- |
| | | | BLUE-4↑ | CIDEr↑ | ROUGE↑ |
| Qwen2-1.5B | ✗ | ✓ | 14.31 | 76.96 | 36.63 |
| | ✓ | ✓ | 13.33(-0.98) | 77.23(+0.27) | 37.10(+0.47) |
| Qwen2-7B | ✗ | ✓ | 14.34 | 79.70 | 38.11 |
| | ✓ | ✓ | 13.67(-0.67) | 81.34(+0.64) | 38.37(+0.26) |

Table 1: **Analysis of performance with/without pre-train stage.** Further analysis of pre-train and SFT stage . (*) denotes performance change compared to model trained without pre-train stage.

Following the training paradigm of (Liu et al., 2023a), we perform one epoch of both pre-training and Supervised Fine-Tuning (SFT) under the same settings as LL3DA. Models are trained on the ScanQA, Nr3D, ScanRefer, and scene alignment from (Hong et al., 2023).

As shown in Table 1, experiments with the Qwen2-7B model reveal no significant improvement but a performance trade-off. Specially, compared to the SFT-only baseline, training with additional pre-train stage results in a 0.67 decrease in the BLEU-4 score, while CIDEr and ROUGE increases by 0.64 and 0.26, respectively.

While pre-train stage is designed to achieve alignment between encoder and LLM, its ineffectiveness suggests that the model may not be adequately utilizing the features provided by the encoder.

| Model | Adaptive | Qwen-1.5B | | | Qwen2-7B | | |
|---|---|---|---|---|---|---|---|
| | capacity | B-4↑ | C↑ | R↑ | B-4↑ | C↑ | R↑ |
| Only LLM w. LoRA | ✗ | 0.00 | 1.08 | 3.34 | 0.08 | 0.96 | 4.62 |
| Full model w. LoRA | ✓ | 12.24 | 75.28 | 38.12 | 13.54 | 81.92 | 38.37 |
| Only LLM | ✓ | 14.31 | 76.96 | 36.63 | 13.11 | 80.34 | 37.37 |
| Full model | ✓ | 13.33 | 77.23 | 37.10 | 13.67 | 81.34 | 38.37 |

Table 4: **Analysis of training pattern on LLM.** To prevent interference during the pre-train stage, models only undergoes SFT training. Results are evaluated on ScanQA.

### 3.3 DOES YOUR PRE-TRAINED 3D ENCODER WORK?

| Model | ScanQA | | |
|---|---|---|---|
| | B-4↑ | C↑ | R↑ |
| *Official LL3DA (Opt-1.3B)* | | | |
| Full model | 13.53 | 76.69 | 37.31 |
| *Encoder Ablation (Qwen2-1.5B)* | | | |
| No encoder | 11.83 | 75.39 | 38.12 |
| Full model | 13.82 | 77.44 | 36.48 |

Table 2: **Analysis of with/without 3D Encoder.** Only the projector is trained following LL3DA.

| Model | Adaptive capacity | Training stage | | ScanQA | | |
|---|---|---|---|---|---|---|
| | | Pre-train | SFT | B-4↑ | C↑ | R↑ |
| *Q-Former projector (Qwen2-1.5B)* | | | | | | |
| Full model | ✓ | ✓ | ✗ | 13.82 | 77.44 | 36.48 |
| *MLP projector (Qwen2-1.5B)* | | | | | | |
| Full model | ✗ | ✓ | ✗ | 0.00 | 0.57 | 1.10 |
| | ✓ | ✗ | ✓ | 7.94 | 59.87 | 32.00 |
| | ✓ | ✓ | ✓ | 13.00 | 67.27 | 32.69 |

Table 3: **Analysis of using different projector.** Compared with Q-Former as projector, training only a two-layer MLP projector does not yield meaningful results.

Experiments in **Supplementary Material F** show that the 3D encoder has the potential to be utilized after the pre-train stage. Therefore, we further evaluate each part of 3D VLM to find out what is the key factor for the final performance.

We first examine the impact of the 3D encoder on the model under the same setting to LL3DA. As demonstrated in Table 2, model without encoder can achieve similar performance to model with encoder. Specially, model without 3D encoder achieves 98% performance on CIDEr and can even improve ROUGE from 36.48 to 38.12, which reveals that 3D encoder does not affect the final performance in this setting. Furthermore, given that only train the projector of a 3D VLM without 3D encoder resulting in such performance, we can infer that models can learn knowledge across multiple scenes solely by relying on the scene-agnostic projector, Qformer. This finding drives us to further analyze the projector.

We next examine how different projector and training stage influence the final performance. As shown in Table 3, only train the two-layer MLP projector during pre-train stage does not yield meaningful results as leveraging Q-Former. However, as long as models with the MLP projector has undergone SFT, it can achieve a certain level of performance on ScanQA.

Considering the results of Table 2 and Table 3 in combination, we attribute this phenomenon to the model's overfitting to the pure-text in Question-Answering, with varying performance depending on whether the trained model possesses adaptive capability. More specially, since the model will be overfitted to pure-text, it is insensitive to the input of the 3D Encoder. There is no need to learn the 3D scene in the Q-Former and it only requires modeling the relationship between the input question and the answer, while only the MLP projector has no such adaptive capacity. Therefore, only train the MLP projector yield almost 0 in CIDEr, but achieve 67.27 in CIDEr due to he adaptive capacity included by LLM during SFT.

To verify this hypothesis, we conduct further analysis on the LLM. To eliminate other factors, we directly train the models with only SFT. As shown in Table 4, we can firstly observe that when trained with SFT, the performance gap between the only LLM setup and the full Model setup is not significant. 3D VLM with only LLM can achieve 99.65% and 98.77% performance on ScanQA CIDEr with Qwen2-1.5B and Qwen2-7B, respectively. However, if LoRA is equipped to the only

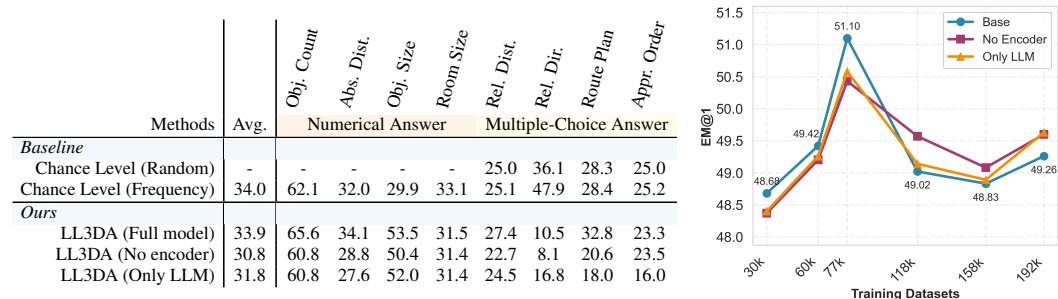

| Methods | Avg. | Obj. Count | Abs. Dist. | Obj. Size | Room Size | Rel. Dist. | Rel. Dir. | Route Plan | Appr. Order |
|---|---|---|---|---|---|---|---|---|---|
| | | Numerical Answer | | | | Multiple-Choice Answer | | | |
| *Baseline* | | | | | | | | | |
| Chance Level (Random) | - | - | - | - | - | 25.0 | 36.1 | 28.3 | 25.0 |
| Chance Level (Frequency) | 34.0 | 62.1 | 32.0 | 29.9 | 33.1 | 25.1 | 47.9 | 28.4 | 25.2 |
| *Ours* | | | | | | | | | |
| LL3DA (Full model) | 33.9 | 65.6 | 34.1 | 53.5 | 31.5 | 27.4 | 10.5 | 32.8 | 23.3 |
| LL3DA (No encoder) | 30.8 | 60.8 | 28.8 | 50.4 | 31.4 | 22.7 | 8.1 | 20.6 | 23.5 |
| LL3DA (Only LLM) | 31.8 | 60.8 | 27.6 | 52.0 | 31.4 | 24.5 | 16.8 | 18.0 | 16.0 |

Table 5: **Evaluation on VSI-bench and SQA3D. Left:** Evaluation on VSI-bench with models trained on datasets from VLM3R. LL3DA adopts Qwen2-7B as LLM backbone. **Right:** Evaluation on SQA3D about data scaling and model setup with Qwen2-7B.

| Method | ScanQA | | | SQA3D | |
|---|---|---|---|---|---|
| | BLUE-4↑ | CIDEr↑ | EM↑ | EM↑ | EM-R↑ |
| ***3D object-centric 3D VLM*** | | | | | |
| ChatScene reproduction (7B) | 11.0 | 83.4 | 19.8 | 54.2 | 56.8 |
| No Encoder | 13.0(+2.0) | 83.8(+0.4) | 19.6(-0.2) | 54.1(-0.1) | 56.6(-0.2) |
| Only LLM | 9.8(-3.2) | 68.3(-15.5) | 15.4(-4.2) | 52.2(-1.9) | 52.2 (-4.4) |
| ***2D image-based 3D VLM*** | | | | | |
| Video-3D-LLM reproduction (7B) | 14.2 | 100.7 | 29.7 | 58.3 | 62.1 |
| No Encoder | 11.1(-3.1) | 79.6(-21.1) | 24.1(-5.6) | 51.8(-6.5) | 54.1(-8.0) |
| Only LLM | 10.0(-1.1) | 76.0(-3.6) | 21.7(-2.4) | 51.4(-0.4) | 53.8(-0.3) |

Table 6: **Evaluation with 3D object-centric and 2D image-based 3D VLMs.** (*) indicate performance change compared to results before the ablation step. Results shows that 3D object-centric VLM also has lower encoder utilization, while 2D image-based VLM has higher encoder utilization.

LLM setup in SFT, the performance will immediately drop to almost 0 due to the insufficient adaptive capacity provided by LoRA, which also aligns with our findings.

## 4 FURTHER VERIFICATION

### 4.1 EXPERIMENTS ON VSI-BENCH AND SQA3D

To further validate our findings, we conduct additional ablation studies on both VSI-bench(Yang et al., 2025) and SQA3D(Ma et al., 2022) to assess the model's reliance on the encoder and projector. Given the model's limited instruction-following capability on VSI-bench, we perform SFT using the dataset proposed in VLM3R (Fan et al., 2025). Similarly, we also compare the performance under three setups: with the full model, with only the 3D encoder removed, and with the only LLM.

As shown in the left part of Table 5, the results on VSI-bench indicate that the model's accuracy decreases only marginally when removing encoder. Without the encoder, the performance of LL3DA only drops from 33.9 to 30.8. More specially, under no encoder and only LLM setup, model can achieve 90.85% and 93.85% performance of full model setup, respectively.

As shown in the right part of Table 5, results on SQA3D reveal that ablations on model and data scale have little impact on performance. More specially, as the dataset scale and model setup vary, the performance on SQA3D fluctuates between 48.69 and 51.10. When the dataset size used goes to 77k, the model performance reaches a peak of 51.10. At this point, the no encoder and only LLM setups can achieve 98.68% and 98.98% performance of the full model setup, respectively.

### 4.2 EXPERIMENTS WITH MORE APPROACHES

To further validate our findings, we conduct additional ablation studies with other methods, specifically ChatScene and Video-3D-LLM on the ScanQA and SQA3D. As shown in Table 6, results for ChatScene, representing 3D object-centric 3D VLM, show minimal performance degradation when the 3D object encoder is ablated. Furthermore, its performance on CiDER and BLUE-4 even shows a

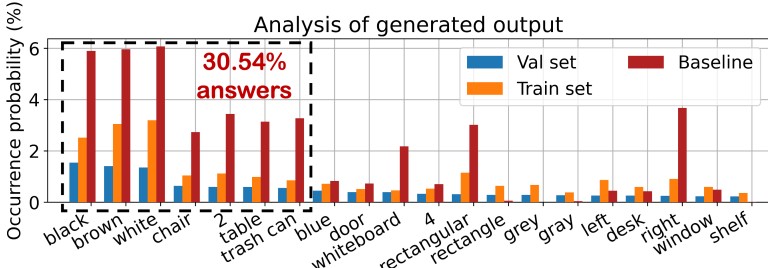

Figure 3: **Analysis of generated answer frequency.** Distribution of the top 20 generated answers, which shows that model is overfitting to the frequent answers. Baseline denotes LL3DA with Qwen2-7B.

Table 7: **Predicted answer percentage by top $k$ frequent answer.**

| | Top $k$ answer | Answer percent |
|---|---|---|
| Val set | 7 | 6.69 |
| Train set | 7 | 12.78 |
| Baseline | 7 | 30.54 |
| Val set | 20 | 10.68 |
| Train set | 20 | 21.21 |
| Baseline | 20 | 43.18 |

| Method | Without balancing | | | With balancing | | |
|---|---|---|---|---|---|---|
| | BLUE-4↑ | CIDEr↑ | ROUGE↑ | BLUE-4↑ | CIDEr↑ | ROUGE↑ |
| Only LLM | 13.11 | 80.34 | 37.37 | 16.32(+3.21) | 89.17(+8.83) | 39.97(+2.60) |
| Full model | 13.67 | 81.43 | 38.37 | 14.67(+1.00) | 84.23(+2.80) | 38.42(+0.05) |

Table 8: **Effectiveness of data balancing on ScanQA.** Simple data balancing improve the overall performance but still allows the model to learn shortcuts. (*) indicates the performance change compared to without balancing.

slight improvement. This suggests that the encoder is similarly insignificant for ChatScene, while a significant drop in performance is observed on ChatScene only LLM setup. Further analysis please refer to **Supplementary Material I.2**, which reveals that this performance drop is primarily due to the ablation of 2D scene image features, but not the 3D information.

Conversely, ablating the 2D image encoder from the 2D image-based method, Video-3D-LLM, led to a substantial performance decrease. Specially, CIDEr on ScanQA decreases by 21.1, and EM on SQA3D drops by 8, which suggests that Video-3D-LLM utilize their encoder inputs more effectively.

In summary, our experiments with ChatScene and Video-3D-LLM support the observation by (Mo & Liu, 2024) that 3D-centric methods can overfit to training data. However, our findings further reveal that this overfitting may extend to the textual descriptions alone, suggesting that the 3D encoder is not a crucial component for these methods.

## 5 METHOD & EXPERIMENTS

After confirming that the issue does not stem from data format difference, specifically data of 3D VLM divergence from the common multiple-choice format used in 2D VLMs (as detailed in **Supplementary Material G**), our analysis identifies the data's underlying distributional properties and training characteristics in 3D VLMs as the main source of the problem. In the subsequent sections, we will first examine how distributional imbalance contributes to overfitting. We then demonstrate that while data balancing of under-sampling improves overall performance, it does not fully resolve the issue. Finally, we introduce our proposed sequence rearrangement, to mitigate this persistent challenge.

### 5.1 ANSWER DISTRIBUTION

To demonstrate how the model overfits the most frequent answers in the long-tailed ScanQA, we analyze the distribution of its predicted answers. The following analysis compares the model's predictions to the ground-truth distribution of train and val set, revealing a severe concentration on a few high-frequency answers.

As illustrated in Fig. 3, we visualize the answer occurrence probabilities. We sort the answers based on their frequency in the val set with visualization of the predicted answer distributions. It is evident that on ScanQA, the predicted answers are significantly concentrated on the most frequent answers, with a probability much higher than their occurrence in the train and val set. Furthermore, results in Table 7 reveal that only top 7 frequent answer (within the black dashed box) and top 20 frequent answer accounts for 30% and 43.18% of the entire predicted answers, respectively.

| Method | Training stage | | without balancing | | | with balancing | | |
|---|---|---|---|---|---|---|---|---|
| | Pre-train | SFT | BLUE-4↑ | CIDEr↑ | ROUGE↑ | BLUE-4↑ | CIDEr↑ | ROUGE↑ |
| Only LLM | ✗ | ✓ | 0.12 | 5.81 | 6.80 | 0.15 | 4.50 | 3.49 |
| Full model | ✓ | ✗ | 0.00 | 2.03 | 4.60 | 0.00 | 1.82 | 7.11 |
| Full model | ✗ | ✓ | 8.63 | 69.19 | 35.00 | 9.47 | 65.56 | 33.89 |
| Full model | ✓ | ✓ | 9.41 | 66.55 | 34.67 | 11.69 | 72.38 | 34.07 |

Table 9: **Experiments with sequence rearrangement on ScanQA.** Adopting Qwen2-7B as LLM backbone, sequence rearrangement prevents the model from taking shortcuts with textual context, thereby encouraging it to learn genuine 3D spatial understanding .

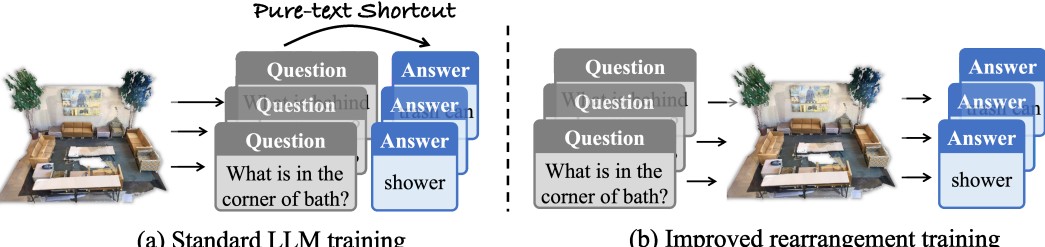

(a) Standard LLM training      (b) Improved rearrangement training

Figure 4: **Visualization of sequence rearrangement.** By simply swapping the positions of 3D scene and quesiton, sequence rearrangement can break the text shortcut.

## 5.2 DATA BALANCING

Inspired by (Goyal et al., 2017), we apply under-sampling to balance the ScanQA dataset, setting a threshold of 50 for the number of questions per answer. As shown in Table 8, this approach successfully improve the model's overall performance. However, we observe that the model's reliance on the encoder remain unchanged. More specifically, following the data balancing procedure, the only LLM setup continue to demonstrate an ability to answer 3D scene questions by relying solely on the textual context. Notably, its performance of CIDEr on the ScanQA, is even improved from 84.23 to 87.94, surpassing the full model setup. This result suggests that although data balancing help to alleviate some dataset biases, it is insufficient to force the model to genuinely incorporate and learn from the 3D scene information.

## 5.3 SEQUENCE REARRANGEMENT

A key difference between 3D scene understanding and 2D visual question answering is that a single scene can be correspond to multiple QA pairs. When a scene has numerous annotations with imbalance, the primary scene input may become irrelevant prefix, while modeling question-answer relationship with text comprehension of LLM. Therefore, we further leverage sequence rearrangement to encourage model for understanding 3D scene.

As illustrated in Fig. 4, we rearrange the input sequence to address this issue. Specifically, let $\mathbf{S}$, $\mathbf{Q}$ and $\mathbf{A}$ denote the 3D **S**cene, **Q**uestion, and **A**nswer, respectively. 3D VLMs are trained to learn the distribution $P(\mathbf{A}|\mathbf{S}, \mathbf{Q})$ through next-token prediction. However, due to training pattern and data imbalance, it tends to learn a shortcut by only modeling pure-text Question-Answering sentence $[\mathbf{Q}, \mathbf{A}]$. Our sequence rearrangement changes the input order from $[\mathbf{S}, \mathbf{Q}, \mathbf{A}]$ to $[\mathbf{Q}, \mathbf{S}, \mathbf{A}]$. To leverage the powerful comprehension and reasoning capabilities of LLMs, the model must successfully map the $\mathbf{S}$ to the text latent space through pre-train stage and understand it accordingly. This change prevents the model from relying only on its language understanding to model $P(\mathbf{A}|\mathbf{Q})$ and forces it to engage in genuine 3D scene understanding.

As shown in Table 9, results demonstrate the effectiveness of this rearrangement. Experiments show that setups with only LLM or only pre-train stage perform almost 0 on ScanQA. This aligns with our expectations, as the LLM-only model cannot process the 3D scene, while only pre-train the model lacks instruction-following capabilities on ScanQA. It is worth noting that without data balancing, the contribution of the pre-train stage is previously difficult to isolate in ablation studies.

| Methods | Avg. | Obj. Count | Abs. Dist. | Obj. Size | Room Size | Rel. Dist. | Rel. Dir. | Route Plan | Appr. Order |
|---|---|---|---|---|---|---|---|---|---|
| | | Numerical Answer | | | | Multiple-Choice Answer | | | |
| *3D scene-centric* | | | | | | | | | |
| LL3DA | 33.9 | 65.6 | 34.1 | 53.5 | 31.5 | 27.4 | 10.5 | 32.8 | 23.3 |
| LL3DA w/ rearrangement | 41.0 | 65.6 | 33.77 | 54.3 | 32.3 | 36.6 | 45.6 | 32.8 | 24.1 |
| *2D image-based* | | | | | | | | | |
| VGLLM-4B reproduction | 46.1 | 66.4 | 36.6 | 55.2 | 56.3 | 40.8 | 43.4 | 30.4 | 39.5 |
| VGLLM-4B w/ rearrangement | 48.2 | 67.0 | 33.7 | 54.1 | 61.2 | 44.2 | 44.1 | 29.9 | 51.0 |

Table 10: **Comparisons with 2D image-based approaches.** Comparisons on VSI-bench with our proposed rearrangement. Models with rearrangement achieve better performance.

| Method | Balancing | | | Rearrangement | | | Rearrange. & balancing | | |
|---|---|---|---|---|---|---|---|---|---|
| | B-4↑ | C↑ | R↑ | B-4↑ | C↑ | R↑ | B-4↑ | C↑ | R↑ |
| ChatScene | 10.81 | 80.42 | 39.43 | 6.56 | 60.84 | 31.89 | 8.69 | 67.79 | 32.16 |
| Ours | **14.67** | **84.23** | **38.42** | **9.41** | **66.55** | **34.67** | **11.69** | **72.38** | **34.07** |

Table 11: **Comparisons to 3D object-centric approach on ScanQA.**

However, by combining data balancing with sequence rearrangement, our ablation study clearly reveals the distinct contributions of pre-train and SFT, which is consistent with established findings in 2D VLMs. More specially, only LLM setup achieve only 6.21% performance of full model setup on CIDEr, while pre-train stage clearly improve performance from 65.56 to 72.38.

## 5.4 COMPARISONS TO STATE-OF-THE-ART APPROACHES

**Comparison with 2D image-based approaches.** 2D image-based approaches typically leverage superior visual encoders for robust feature extraction. To evaluate the potential of our method in this context, we apply the proposed rearrangement strategy to the state-of-the-art 2D baseline, VGLLM (Zheng et al., 2025a). As shown in Table 10, this integration yields a performance gain of 2.0 points, improving from 46.1 to 48.1. This demonstrates that our rearrangement mechanism provides critical spatial cues that effectively complement powerful 2D features.

**Comparison with 3D object-centric approaches.** To ensure a fair comparison, we exclude the 2D image features from ChatScene to strictly evaluate 3D-native understanding. As shown in Table 11, our method consistently outperforms this object-centric counterpart on ScanQA. Notably, our approach achieves superior performance even without data balancing, confirming that our scene-centric formulation captures global context more effectively than object-centric approaches.

## 6 CONCLUSION

This work reveals a fundamental flaw in 3D scene-centric models: their success often stems from learning data shortcuts that bypass the 3D encoder, rather than from genuine scene understanding. This reliance on textual cues explains the previously observed minimal impact of dataset scaling and pre-training. To address this, we introduce a simple input sequence rearrangement that disrupts these shortcuts and compels the model to ground its reasoning in the 3D scene. Our experiments confirm that this approach restores the essential role of the 3D encoder, providing a solid foundation for building and evaluating models with true spatial intelligence.

**Limitations.** While our method establishes a more robust foundation for genuine 3D scene understanding, we acknowledge that its performance on standard benchmarks currently lags behind the state-of-the-art 2D image-based approaches. We hypothesize that previous top-performing 3D VLMs achieve their results in part by exploiting the shortcuts we have identified. Future work should focus on more fair comparison and performance improvement with real 3D scene understanding.

**Reproducibility Statement**

To ensure the reproducibility of our research, we provide the complete source code, including implementation, training, and evaluation scripts in the supplementary material. All experiments are conducted on publicly available datasets as described in the main paper. Detailed descriptions of our data processing pipeline, particularly the under-sampling method used for data balancing, are available in appendix. Furthermore, appendix documents all hyperparameters used for our experiments, and specifies the computing environment. We believe the combination of our paper, appendix, and the provided code contains all the necessary details to reproduce our findings.

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

## A    APPENDIX/SUPPLEMENTAL MATERIAL

The outline of the Appendix is as follows:

- More implementation details;
- More analysis on data scaling capabilities on 3D-QA;
    - Additional analysis on SQA3D;
    - Additional analysis on VSI-bench;
- More analysis on data scaling capabilities on 3D-DC;
    - Additional analysis on ScanRefer;
    - Additional analysis on Nr3D;
    - Additional analysis on data isolation;
- More analysis on data scaling capabilities on 3D-VG;
    - Grounding representation;
    - Additional analysis on ScanRefer;
- More analysis on encoder alignment;
    - Additional visualization of token distribution;
    - Additional analysis of semantic information;
- More analysis on data format;
    - Additional analysis of question template;
    - Additional analysis of ScanQA-Choice;
- More analysis on data balancing;
    - Additional ablation study of data balancing;
    - Additional ablation study on model with data balancing;
    - Additional analysis on comparisons to 3D scene-centric methods;
- More discussion;
    - Discussion on performance analysis of 3D scene-centric methods;
    - Discussion on performance analysis of 3D object-centric methods;
    - Discussion on performance analysis of 2D image-based methods;
    - Discussion on LLM usage;
    - Discussion on social impact;

## B    IMPLEMENTATION DETAILS

**Datasets.** Following  (Chen et al., 2024a), we utilize the ScanNet dataset (Dai et al., 2017), which comprises 1,201 and 312 diverse and complex indoor 3D scenes for training and validation, respectively. By default, experiments are conducted with the same setting with  (Chen et al., 2024a) no ScanQA (Azuma et al., 2022), ScanRefer (Chen et al., 2020), Nr3D (Achlioptas et al., 2020) and the ScanNet subset of 3D-LLM (Hong et al., 2023). We further divide the ScanNet subset of 3D-LLM into two parts: 3D-LLM QA and 3D-LLM Pre. The 3D-LLM Pre subset encompasses scene descriptions, conversations, and embodied planning tasks. We further leverage Multi3DRefer (Zhang et al., 2023b), Scan2Cap (Chen et al., 2021), 3RScanQA and scene alignment from LEO (Huang et al., 2023b).

**Metrics.** We adopt C, B-4, R as abbreviations for CIDEr (Vedantam et al.), BLEU-4 (Papineni et al., 2002), and Rouge-L (Lin, 2004) to evaluate the quality of the generated textual responses, while accuracy and EM@1 is leveraged to further evaluate quality.

**Implementation Details.** We follow (Chen et al., 2024a) to sample 40k point clouds from each scene for 3D scene encoder (Chen et al., 2024b). We leverage open-source Qwen2-1.5B and Qwen2-7B (Yang et al., 2024) as LLM backbone and Q-Former (Li et al., 2023) as projector by default. Following  (Liu et al., 2023a), we train model by pre-train and SFT stages for 1 epoch, respectively.

We adopt AdamW (Loshchilov & Hutter, 2017) as optimizer with a weight decay of 0.1 and a learning rate decaying from $10^{-4}$ to $10^{-5}$ with a cosine annealing scheduler for pre-train stage, while a learning rate decaying from $5\times10^{-5}$ to $10^{-5}$ is leveraged for SFT stage. Seed is set to 0 for all experiments. For all the training tasks, we train with a total batch size of 32 on 8×Ascend-D910 (64G) NPU. The max count of data balancing is set to 50 for the best performance. We observe that training for only one epoch in SFT stages without gradient clipping, can yield comparable performance to that achieved by LL3DA training Q-Former for 32 epochs.

## C  ADDITIONAL ANALYSIS OF DATA SCALING CAPABILITIES ON 3D-QA

### C.1  ADDITIONAL ANALYSIS ON SCANQA

| LLM | Dataset | ScanQA | | |
|---|---|---|---|---|
| | | BLUE-4 ↑ | CIDEr↑ | ROUGE ↑ |
| Qwen2-1.5B | 30k | 11.12 | 70.06 | 36.47 |
| | 64k | 10.90 | 70.88 | 36.00 |
| | 105k | 12.46 | 73.49 | 36.82 |
| | 145k | 13.33 | 77.23 | 37.10 |
| | 162k | 13.63 | 77.38 | 36.80 |
| | 263k | 12.87 | 76.42 | 36.56 |
| | 355k | 13.64 | 78.56 | 37.44 |
| | 661k | 12.65 | 77.31 | 37.43 |
| Qwen2-7B | 30k | 10.61 | 78.69 | 38.93 |
| | 145k | 13.67 | 81.34 | 38.37 |
| | 661k | 14.43 | 81.52 | 38.57 |

Table 12: **Analysis of large-scale scaling capabilities.** Gray lines represents the result after adding alignment data only for pre-train.

| Methods | CIDER↑ |
|---|---|
| ***2D image-based 3D VLM*** | |
| LLaVA-3D | 91.7 |
| Video-3D-LLM | 102.1 |
| ***3D object-centric 3D VLM*** | |
| ChatScene | 87.7 |
| ***3D scene-centric 3D VLM*** | |
| Ours-1.5B(661k) | 77.31 |
| Ours-1.5B(best) | 79.56 |
| Ours-7B(661k) | 81.52 |

Table 13: **Comparisons on ScanQA with CIDEr.** Still a great gap to other methods.

We first analyze the scaling capabilities of 3D VLM on ScanQA. Further analysis of data scaling capabilities on more 3D tasks can be found in **Supplementary Material C,D,E**, which aligns with the findings on unstable data scaling capabilities of 3D VLM.

As shown in Table 12, progressively increasing the data scale leads to an unstable improvement in performance. Furthermore, switching to larger models results in improvement under the same dataset. More specially, scaling data from 145k to 661k, CIDEr only increases from 81.34 to 81.52 with Qwen2-7B, while CIDEr increases from 77.31 to 81.52 after switching to lager model.

Moreover, as shown in Table 13, after up scaling dataset to 661k, there is still a significant performance gap between our results and other approaches. Specially, official ChatScene and Video-3D-LLM outperform with 6.18 and 20.58 on CIDEr, respectively.

Particularly in the gray line of Table 12, we observe that the increased alignment data during the pre-train stage provides almost no improvement. After adding 306k of alignment data to form the final 661k dataset, the overall performance of the model even declines. This prompted us to further examine the pre-train stage of the model.

### C.2  ADDITIONAL ANALYSIS ON SQA3D

As shown in Table 14, our further experiments on SQA3D to validate data scaling show that SQA3D is largely insensitive to additional dataset or model setups, regardless of its scale or task. Increasing the amount of training data has minimal impact on the model's performance on SQA3D. This observation will be further analyzed in **Supplementary Material H**.

### C.3  ADDITIONAL ANALYSIS ON VSI-BENCH

As shown in Table 15, we also conduct data scaling experiments on the VSI-bench. Unlike ScanQA and SQA3D, VSI-bench includes both numerical and multiple-choice answers, which differ significantly in distribution. Our experiments reveals that the multiple-choice answers are the most

| Dataset | Full model | No encoder weight | No encoder | Only llm | Update |
|---------|-----------|-------------------|-----------|----------|--------|
| 30k | 48.68 | 48.04 | 48.37 | 48.40 | SQA3D (3D-QA) |
| 60k | 49.42 | 49.26 | 49.20 | 49.26 | further +ScanQA (3D-QA) |
| 77k | 51.10 | 50.67 | 50.43 | 50.58 | further +3D-LLM QA (3D-QA) |
| 117k | 49.02 | 49.29 | 49.57 | 49.14 | further +Nr3D (3D-DC) |
| 158k | 48.83 | 48.86 | 49.08 | 48.89 | further +ScanRefer (3D-DC) |
| 192k | 49.26 | 49.69 | 49.60 | 49.63 | further +3D-LLM Pre (Alignment) |

Table 14: **Further analysis of data scaling capabilities on SQA3D.** Results show that the performance of SQA3D is minimally affected by the scale of the training data.

| Dataset | Total Avg. | Numerical Answer Avg. | Multiple-Choice Answer Avg. | Update |
|---------|-----------|-----------------------|-----------------------------|--------|
| 53k | 33.92 | 20.73 | 47.63 | VLM3R Training set |
| 83k | 23.03 | 0.00 | 46.79 | further +ScanQA (3D-QA) |
| 113k | 20.36 | 0.06 | 34.36 | further +SQA3D (3D-QA) |
| 130k | 29.82 | 11.93 | 48.43 | further +3D-LLM QA (3D-QA) |
| 170k | 23.24 | 0.00 | 47.14 | further +Nr3D (3D-DC) |
| 211k | 24.04 | 1.20 | 47.78 | further +ScanRefer (3D-DC) |
| 245k | 25.85 | 3.50 | 49.11 | further +3D-LLM Pre (Alignment) |

Table 15: **Further analysis of data scaling capabilities on VSI-bench.** Results shows that data scaling with other datasets negatively impacts the model's instruction-following ability on multiple-chocie questions.

affected, with the model losing nearly all performance on these questions. We find that training with additional data negatively impacted the model's instruction-following ability, causing it to output answers directly instead of the required multiple-choice options.

# D   ADDITIONAL ANALYSIS OF DATA SCALING CAPABILITIES ON 3D-DC

| LLM | 3D-DC | | 3D-QA | Alignment | BLUE-4 ↑ | CIDEr↑ | ROUGE ↑ |
|-----|-------|--------|-------|-----------|----------|--------|---------|
| | Nr3D | ScanRefer | ScanQA | 3D-LLM | | | |
| *Official LL3DA* | | | | | | | |
| Opt-1.3B | ✓ | ✓ | ✓ | ✓ | 13.37 | 23.94 | 45.78 |
| *Scaling with other task* | | | | | | | |
| | ✓ | ✗ | ✗ | ✗ | 23.12 | 38.58 | 52.00 |
| Qwen2-1.5B | ✓ | ✗ | ✓ | ✗ | 22.89(-0.23) | 36.86(-1.72) | 51.29(-0.71) |
| | ✓ | ✗ | ✓ | ✓ | 23.62(-0.40) | 39.12(+0.54) | 51.69(-0.31) |
| *Scaling with 3D-DC* | | | | | | | |
| | ✓ | ✗ | ✗ | ✗ | 23.12 | 38.58 | 52.00 |
| | ✓ | ✓ | ✗ | ✗ | 12.05(-11.07) | 18.23(-20.35) | 42.94(-9.06) |
| Qwen2-1.5B | ✓ | ✓ | ✓ | ✗ | 11.55(-11.57) | 18.21(-20.37) | 42.88(-9.12) |
| | ✓ | ✓ | ✓ | ✓ | 11.64(-11.48) | 18.29(-20.29) | 42.54(-9.46) |

Table 16: **Further analysis of data scaling capabilities on Nr3D.** Following our investigation into the data scaling capabilities for the 3D Dense Captioning task with Nr3D-centric setting, our further analysis reveals that Nr3D does not benefit from data scaling. On the contrary, it potentially leads to a degradation in performance. 3D-LLM denotes the scene-alignment dataset (Hong et al., 2023). (*) denotes performance change compared to train only on Nr3D.

| LLM | 3D-DC | | 3D-QA | Alignment | BLUE-4 ↑ | CIDEr ↑ | ROUGE ↑ |
|---|---|---|---|---|---|---|---|
| | ScanRefer | Nr3D | ScanQA | 3D-LLM | | | |
| *Official LL3DA* | | | | | | | |
| Opt-1.3B | ✓ | ✓ | ✓ | ✓ | 35.97 | 62.98 | 54.65 |
| *Scaling with other task* | | | | | | | |
| Qwen2-1.5B | ✓ | ✗ | ✗ | ✗ | 32.42 | 53.57 | 50.84 |
| | ✓ | ✗ | ✓ | ✗ | 33.60(+1.18) | 54.51(+0.94) | 51.33(+0.49) |
| | ✓ | ✗ | ✓ | ✓ | 33.24(+0.82) | 56.51(+2.94) | 51.00(+0.16) |
| *Scaling with 3D-DC* | | | | | | | |
| Qwen2-1.5B | ✓ | ✗ | ✗ | ✗ | 32.42 | 53.57 | 50.84 |
| | ✓ | ✓ | ✗ | ✗ | 33.62(+1.12) | 54.51(+0.94) | 51.55(+0.71) |
| | ✓ | ✓ | ✓ | ✗ | 33.00(+0.58) | 55.13(+1.56) | 51.22(+0.38) |
| | ✓ | ✓ | ✓ | ✓ | 33.26(+0.84) | 56.16(+2.58) | 51.31(+0.47) |

Table 17: **Further analysis of data scaling capabilities on ScanRefer.** Following our investigation into the data scaling capabilities for the 3D-DC task with ScanRefer-centric setting, our further analysis reveals that ScanRefer benefits from data scaling with pre-train dataset, 3D-QA and 3D-DC datasets. 3D-LLM Pre denotes the scene-alignment dataset (Hong et al., 2023). (*) denotes performance change compared to train only on ScanRefer.

| LLM | Role isolation | | Nr3D | | | ScanRefer | | |
|---|---|---|---|---|---|---|---|---|
| | QA template | Prompt prefix | B-4 ↑ | C↑ | R ↑ | B-4 ↑ | C↑ | R ↑ |
| Qwen2-1.5B | ✗ | ✗ | 12.05 | 18.23 | 42.94 | 33.62 | 54.51 | 51.55 |
| | ✓ | ✗ | 17.79 | 31.92 | 46.92 | 30.59 | 51.89 | 49.20 |
| | ✓ | ✓ | 17.54 | 28.79 | 46.70 | 31.23 | 52.62 | 49.44 |

Table 18: **Analysis of role isolation for 3D VLM.** Further investigation involves the integration of isolation mechanisms to address potential conflicts observed in the Nr3D and ScanRefer datasets. While the implementation of this technique facilitates a more balanced performance profile across the two datasets, it does not ultimately yield peak performance in either individual evaluation.

## D.1 ADDITIONAL ANALYSIS ON NR3D

As shown in Table 16 and Table 17, we further supplement analysis of data scaling capabilities on 3D Dense Captioning task. As shown in Table 16, we observe that Nr3D does not benefit from data scaling from other tasks, nor does it experience a degradation in its original performance. However, we find a catastrophic performance decline on Nr3D when incorporating the ScanRefer dataset, which also focuses on 3D-DC in LL3DA.

## D.2 ADDITIONAL ANALYSIS ON SCANREFER

Analysis of the model's generated outputs during evaluation reveals that ScanRefer contains a high frequency of similar location descriptions starting with "it is to the". This prevalent phrase leads the model to generate such descriptions even on the Nr3D dataset, consequently impacting performance. This observation aligns with our findings regarding data distribution discussed in Section 5.1. However, as shown in Table 17, ScanRefer-centric analysis yields a contrasting conclusion: ScanRefer demonstrates effective data scaling, showing performance improvements across both similar and dissimilar tasks.

### D.3 ADDITIONAL ANALYSIS ON DATA ISOLATION

As shown in Table 18, to further investigate whether role isolation could mitigate the conflicts between datasets, we explore adding dataset-specific prefixes to questions and using distinct question templates for each dataset. While this approach offers some relief, we observe that the performance after role isolation consistently ends up being worse than the best performance achieved on the original datasets separately. Thus, role isolation appears to represent a trade-off rather than a definitive solution.

## E ADDITIONAL ANALYSIS OF DATA SCALING CAPABILITIES ON 3D-VG

### E.1 GROUNDING REPRESENTATION

To facilitate better learning of 3D-VG, we represent each 3D bounding box as $[x, y, z, w, h, l]$, where $x$, $y$ and $z$ denote the coordinate of object center on x-axis, y-axis and z-axis respectively and $w$, $h$ and $l$ denote the width, height and length of the 3D bounding boxes respectively. Let $x_{min}, y_{min}, z_{min}$ represent the minimum value of 3D scene point clouds on x-axis, y-axis and z-axis respectively, and $x_{max}, y_{max}, z_{max}$ represent the maximum value of 3D scene point clouds on x-axis, y-axis and z-axis respectively. We normalize the object 3D bounding boxes $[x, y, z, w, h, l]$ based on the input scene:

$$x = \frac{x - x_{min}}{x_{max} - x_{min}} \times g, \ y = \frac{y - y_{min}}{y_{max} - y_{min}} \times g, \ z = \frac{z - z_{min}}{z_{max} - z_{min}} \times g, \quad (1)$$

where $g$ denotes the maximum value of normalized grid, which is set to 255.

Similarly, we can normalize the 3D bounding box sizes $(w,h,l)$. Considering that the minimum possible value for a 3D bounding box size is zero, we explore two normalization approaches:

**Signed Normalization:** $w = \frac{w - x_{min}}{x_{max} - x_{min}} \times g, \ h = \frac{h - y_{min}}{y_{max} - y_{min}} \times g, \ l = \frac{l - z_{min}}{z_{max} - z_{min}} \times g \quad (2)$

**Min-zero Normalization:** $w = \frac{w}{x_{max} - x_{min}} \times g, \ h = \frac{h}{y_{max} - y_{min}} \times g, \ l = \frac{l}{z_{max} - z_{min}} \times g$

$$(3)$$

### E.2 ADDITIONAL ANALYSIS ON SCANREFER

As shown in Table 19, we further conduct in-depth experiments on 3D-VG. Results indicate that the current performance is comparable to that reported in 3D-LLM with Min-zero Normalization, without considering differences in data scale and model architecture. However, when we use Signed Normalization, model demonstrate failing to learn any meaningful 3D-VG knowledge.

Intuitively, Min-zero Normalization should provide more accurate results. However, the near-zero ACC@0.25 indicates a lack of spatial awareness learned from the 3D scene, consistent with our previous observations. Furthermore, while Signed Normalization on 3D bounding box size yields relatively good performance with larger bounding box sizes after normalization, it suggests that the model's performance might stem from encompassing a wider region through box sizes, rather than precise spatial understanding.

Overall, our findings suggest that without specific architectural designs, it is challenging for general 3D scene-centric VLMs to learn fine-grained spatial information, leading to inaccurate visual grounding.

| LLM | Acc@0.25 ↑ | Dataset | Update |
|---|---|---|---|
| ***Official 3D-LLM*** | | | |
| flamingo | 21.2 | 675k | |
| BLIP2-opt | 29.6 | 675k | |
| BLIP2-flanT5 | 30.3 | 675k | |
| ***Min-zero Normalization*** | | | |
| | 21.8 | 36k | 3D-VG ScanRefer |
| Qwen2-1.5B | 25.8 | 72k | further +3D-DC ScanRefer |
| | 26.2 | 693k | further +full set |
| ***Signed Normalization*** | | | |
| | 1.93 | 36k | 3D-VG ScanRefer |
| Qwen2-1.5B | 2.04 | 72k | further +3D-DC ScanRefer |
| | 2.36 | 693k | further +full set |

Table 19: **Comparisons to 3D-LLM on 3D-VG.** We train model 1 and 4 epoch for pre-train and SFT stages, respectively.

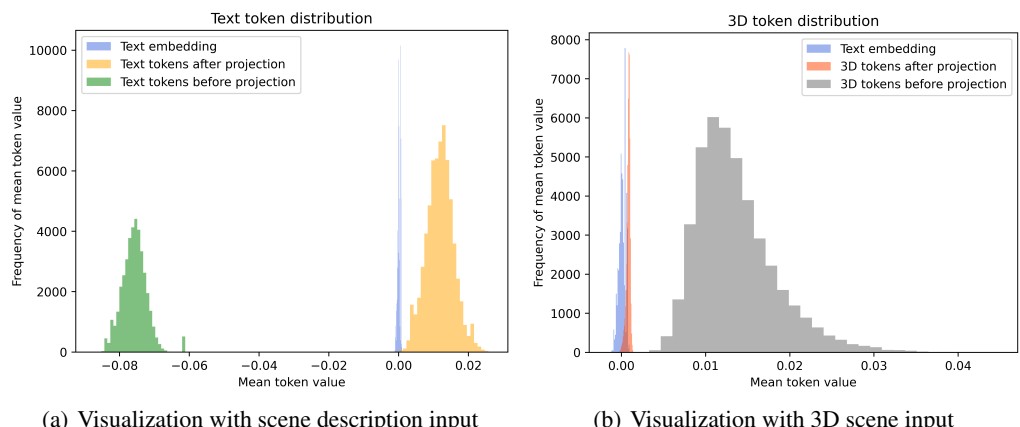

(a) Visualization with scene description input  (b) Visualization with 3D scene input

Figure 5: **Visualization of token distribution with different cross-modal input.** We further visualize the token distribution before and after MLP projector to intuitively express the impact of pre-train stage.

## F   MORE ANALYSIS ON ENCODER ALIGNMENT

### F.1   ADDITIONAL VISUALIZATION OF TOKEN DISTRIBUTION

To better understand the distribution of tokens for alignment analysis on pre-train stage, we collect tokens from the ScanQA training set both before and after the projector, compared to tokens of text embeddings. For each token, we calculate its mean vector and then visualized the distribution of these mean vectors using histograms. This allows for a comparison of how the projector influences the token representations.

As shown in Fig. 5, we further visualize the distribution of text tokens from scene descriptions with CLIP and 3D tokens from the 3D encoder before and after the pre-train stage. While pre-training aims to align disparate data distributions with text for better feature learning, our visualization surprisingly shows that the 3D encoder effectively maps 3D tokens to a distribution even closer to text than using text tokens. This indicates the model's underlying capability to utilize 3D tokens.

| Multi-modal input | Pre-train | ScanQA | | |
|---|---|---|---|---|
| | | BLUE-4 ↑ | CIDEr↑ | ROUGE ↑ |
| ***w/o 3D Object Token*** | | | | |
| Scene description | ScanQA* | 5.18 | 72.42 | 26.68 |
| | 3D-LLM Pre | 5.40 | 75.74 | 27.78 |
| 3D scene | ScanQA* | 5.37(+0.19) | 77.55(+5.13) | 28.35(+1.67) |
| | 3D-LLM Pre | 5.54(+0.14) | 76.30(+0.56) | 27.92(+0.14) |
| ***w/ 3D Object Token*** | | | | |
| Scene description | ScanQA* | 4.85 | 72.67 | 26.84 |
| | 3D-LLM Pre | 5.25 | 75.47 | 27.79 |
| 3D scene | ScanQA* | 5.33(+0.48) | 77.59(+4.92) | 28.44(+1.60) |
| | 3D-LLM Pre | 4.90(-0.35) | 74.40(-1.07) | 27.48(-0.31) |

Table 20: **Analysis of semantic information.** ScanQA* denotes the sampled subset with scene description of ScanQA, and 3D-LLM Pre denotes the scene-alignment dataset (Hong et al., 2023).(*) denotes performance change compared to train with scene description as multi-modal input.

### F.2 ADDITIONAL ANALYSIS OF SEMANTIC INFORMATION

Based on the observations in Section 3.2 and Section 3.3, a straightforward hypothesis is that current 3D scene encoders, often adapted from existing 3D object detection backbones for feature extraction, lack sufficient semantic information compared to 3D object-centric and 2D image-based approaches. Consequently, the pre-train stage alone is insufficient for the LLM to effectively map the extracted 3D features to the text latent space. This limitation potentially leads the model to prioritize learning patterns between questions and answers, rather than achieving genuine visual understanding, thus underutilizing the 3D tokens.

To validate the hypothesis that the 3D encoder lacks sufficient semantic information, we utilize scene descriptions from the ScanNet subset of 3D-LLM (Hong et al., 2023). We encode these descriptions into text embeddings using CLIP to serve as a multi-modal input representing the scene. Given that these descriptions are only available for the ScanNet training split, we sample the final 100 scenes of the train split as a test set and reconstruct the training data for ScanQA, ScanRefer, and Nr3D accordingly. As shown in Table 20, the pre-train stage proves effective when using text embeddings as the 3D scene representation comapred to using ScanQA* as the pre-train data. However, pre-training with the 3D scene encoder tends to be ineffective and can even lead to performance degradation.

Finally, under the same experimental settings, we observe a comparable performance between using text embeddings from scene description and employing 3D tokens extracted by the 3D scene encoder. Despite the potential lack of fine-grained details in the scene descriptions, they still provide information about the object categories and their spatial relationships within the scene. Therefore, in contrast with the initial assumption, we infer that the lack of semantic information in the 3D scene encoder may not be the primary factor contributing to our earlier observations.

As shown in Table 20, our analysis on training augmentations, specifically the inclusion of 3D tokens inside the GT bounding boxes for potential object-level 3D representations, indicate that this augmentation has a minimal impact on the model's performance.

## G MORE ANALYSIS ON DATA FORMAT

### G.1 ADDITIONAL ANALYSIS OF QUESTION TEMPLATE

While current 2D VLMs often evaluate performance on large, well-known datasets using a multiple-choice format, 3D VLMs still primarily rely on traditional metrics. Given the inherent complexity of spatial structures, this raises the question: *would the model be providing a correct understanding that isn't accurately reflected in the evaluation results?* For instance, the answer to "What is in front of

you?" varies depending on a person's orientation in space. In contrast, the multiple-choice format inherently constrains the model's predictions within the distribution of the options.

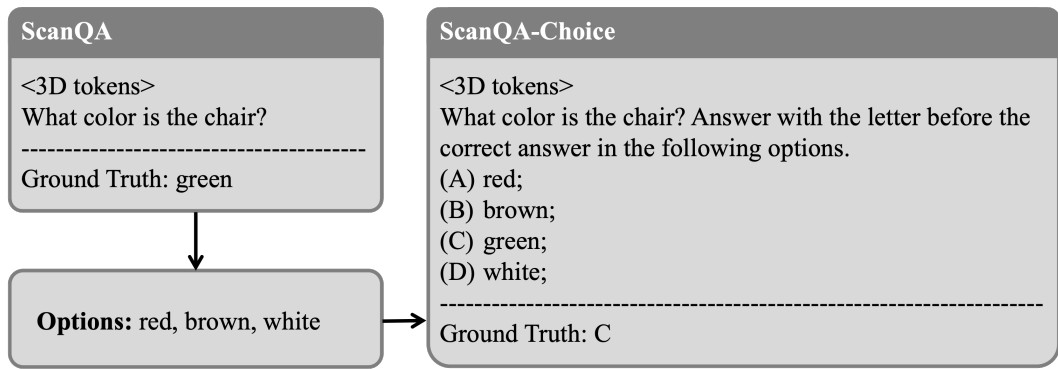

Figure 6: **Example visualization of ScanQA and ScanQA-Choice collection.** Based on the ground truth answer for each question in ScanQA, we sample similar options from the ScanQA answer pool to construct ScanQA-Choice.

To further eliminate the influence of problem format and evaluation metrics, we propose ScanQA-Choice, a multiple-choice version of ScanQA. As shown in Fig. 6, we present a visual demonstration of how ScanQA-Choice is constructed based on ScanQA. We collect the answers from the ScanQA and classify them according to the categories of answers and questions(e.g., quantity, color, object category), assigning the most similar options to each question.

| LLM | 3D input | Dataset | | Accuracy↑ |
| --- | --- | --- | --- | --- |
| | | Pre-train | SFT | |
| *Model scaling* | | | | |
| Opt-125m | ✗ | 3D-LLM Pre | ScanQA | 35.56 |
| Qwen2-0.5B | ✗ | 3D-LLM Pre | ScanQA | 68.15 |
| Qwen2-1.5B | ✗ | 3D-LLM Pre | ScanQA | 88.19 |
| *Training stage* | | | | |
| | ✗ | ScanQA | — | 18.97 |
| Qwen2-1.5B | ✗ | — | ScanQA | 85.26 |
| | ✗ | 3D-LLM Pre | ScanQA | 88.19 |
| Qwen2-1.5B | ✓ | 3D-LLM Pre | ScanQA | 90.65 |
| *Data scaling* | | | | |
| | ✗ | 3D-LLM Pre | $\frac{1}{2}$ScanQA | 86.80 |
| Qwen2-1.5B | ✗ | 3D-LLM Pre | ScanQA | 88.19 |
| | ✗ | 3D-LLM Pre | ScanQA,3D-LLM QA | 91.74 |
| | ✗ | 3D-LLM Pre,ScanQA,3D-LLM QA | ScanQA,3D-LLM QA | 91.70 |

Table 21: **Analysis of experiments on ScanQA-Choice.** 3D-LLM Pre and 3D-LLM QA denotes the scene-alignment and question answering from (Hong et al., 2023). We leverage two-layer MLPs as a projector to avoid Q-Former directly learning the text embedding of the question.

As shown in Table 21, our experiments across various settings reveal clear benefits from model scaling, data scaling, and the pre-train and SFT stages on ScanQA-Choice, which is not evident on the original ScanQA. However, we observe that even without providing 3D token input, the model still achieves high accuracy. This aligns with our findings in Section 5.1, suggesting that the model might be leveraging memorized patterns between questions and answers learned during the SFT stage to attain higher performance. While we attempte to mitigate this by introducing question-irrelevant options in ScanQA-Choice, the overall results remained largely consistent.

## G.2   ADDITIONAL ANALYSIS OF SCANQA-CHOICE

| LLM | 3D input | Dataset | | Accuracy↑ |
| --- | --- | --- | --- | --- |
| | | Pre-train | SFT | |
| **Data scaling** | | | | |
| Qwen2-0.5B | ✗ | 3D-LLM Pre | $\frac{1}{2}$ScanQA | 77.65(-0.9) |
| | | 3D-LLM Pre | ScanQA | 78.55 |
| | | 3D-LLM Pre | ScanQA,3D-LLM QA | 90.48 (+11.93) |
| | | 3D-LLM Pre&QA,ScanQA | ScanQA,3D-LLM QA | 90.93 (+0.45) |
| Qwen2-1.5B | ✗ | 3D-LLM Pre | $\frac{1}{2}$ScanQA | 89.26 (-1.39) |
| | | 3D-LLM Pre | ScanQA | 90.65 |
| | | 3D-LLM Pre | ScanQA,3D-LLM QA | 91.74 (+1.09) |
| | | 3D-LLM Pre&QA,ScanQA | ScanQA,3D-LLM QA | 91.70 (-0.04) |

Table 22: **Further analysis on ScanQA-Choice.** We supplement experiments with Qwen2-0.5B, which can better perform data scaling capabilities. 3D-LLM Pre denotes the scene-alignment dataset (Hong et al., 2023). We leverage two layer MLPs as projector to avoid Q-Former directly learning text embedding of question. (*) denotes performance change compared to the basic setting of leveraging 3D-LLM Pre for pre-training and ScanQA for SFT.

We supplement experiments on ScanQA-Choice with different LLM backbone. As shown in Table 22, the benefits of data scaling are more pronounced when using smaller models, such as Qwen2-0.5B. This suggests that with a larger model like Qwen2-1.5B, performance may have approached saturation on smaller datasets without 3D input.

| Final loss↓ | Accuracy(EM@1)↑ | Ablation step |
| --- | --- | --- |
| 0.3 | 90.82 | Base version of choice ScanQA |
| 0.5 | 84.45 | Delete instructions |
| 0.6 | 75.61 | Delete instructions and option C |
| 0.8 | 65.25 | Delete instructions and option B,C |
| 1.7 | 15.67 | Delete instructions and option A,B,C = Basic ScanQA setting |
| 1.5 | 23.49 | Only provide the length of the answer |
| 1.7 | 17.98 | Randomly sample a question as an option |

Table 23: **Further analysis of instructions with Q-Former on ScanQA-Choice.** 3D VLMs with Qwen2-1.5B and Q-Former are only trained under pre-train stage for 1 epoch. The gray line is equivalent to the original ScanQA, where the Accuracy metric is converted to EM@1. Assuming the correct answer is D among four options of [A,B,C,D].

As shown in Table 23, we supplement further analysis on ScanQA-Choice with Q-Former. When employing Q-Former as the projector, the model, likely due to Q-Former's text processing capabilities, achieves high performance even without the SFT stage.

We further investigated this potential "shortcut" through ablation studies, as shown in Table 24. We observe that instruction prompts are not critical, suggesting the model could inherently learn the relationship between questions and answers. In contrast, the number of answer options prove to be a significant factor. Progressively reducing the number of options led to lower convergence and poorer final performance, with a drastic drop occurring when the last choice was removed.

In summary, our findings confirm that under the current data scale, employing a multiple-choice QA format is not optimal. The model tends to disregard the 3D tokens and instead focuses on learning the newly provided information within the options.

| Final loss↓ | Accuracy(EM@1)↑ | Ablation step |
|---|---|---|
| 0.2 | 89.79 | Base version of choice ScanQA |
| 0.2 | 89.75 | Delete instructions |
| 0.4 | 76.62 | Delete instructions and option C |
| 0.4 | 75.72 | Delete instructions and option B,C |
| 1.5 | 18.56 | Delete instructions and option A,B,C = Basic ScanQA setting |
| 0.2 | 89.6 | No 3D input |

Table 24: **Further analysis of instructions with MLP projector on ScanQA-Choice.** 3D VLMs with Qwen2-1.5B and two MLP layers are trained under pre-train and SFT stages for 1 epoch, respectively. The gray line is equivalent to the original ScanQA, where the Accuracy metric is converted to EM@1. Assuming the correct answer is D among four options of [A,B,C,D].

## H ADDITIONAL ANALYSIS OF DATA BALANCING

### H.1 ABLATION STUDY OF DATA BALANCING

| Balancing | | ScanQA | | | | SQA3D | |
|---|---|---|---|---|---|---|---|
| max count | sample rate | BLUE-4 ↑ | CIDEr↑ | ROUGE ↑ | sample rate | EM |
| *Dataset distribution* | | | | | | |
| - | 100% | 13.67 | 81.43 | 38.37 | 100% | 52.29 |
| *Distribution balancing* | | | | | | |
| 8392 | 100% | 13.67 | 81.43 | 38.37 | 100% | 52.29 |
| 790 | | | | | 58% | 47.87 |
| 100 | 86% | 13.75 | 83.29 | 37.89 | 27% | 39.35 |
| 50 | 78% | 14.67 | 84.23 | 38.42 | 20% | 35.77 |
| 10 | 60% | 14.27 | 84.63 | 40.11 | 9% | 25.21 |
| 1 | 34% | 9.68 | 66.56 | 30.66 | 2% | 22.82 |

Table 25: **Further ablation study on the max count of data balancing.** Leveraging Qwen2-7B as the LLM backbone.

To further investigate the impact of data distribution, we perform data balancing of under-sampling experiments on the ScanQA and SQA3D datasets. Our balancing method first calculates the numbers of question corresponding to each unique answer within the dataset. We then set a maximum number of questions for each answer, denoted as **max count**, and trim surplus questions. The remaining data is then resampled to the original dataset size, while the percentage of original remaining data relative to the original dataset is represented by the **sample rate**.

As shown in Table 25, in ScanQA, the maximum number of questions corresponding to a single answer is 790, while in SQA3D, this number reaches 8392. Our analysis reveals that ScanQA exhibits a severe long-tail distribution, with nearly half of the questions corresponding to answers that appear fewer than 10 times. Consequently, data balancing proves to be highly effective in improving performance on this dataset.

However, a more strictly balanced dataset led to a decrease in performance which shows inherent imbalance in the ScanQA val set. This is because training on a balanced dataset can prevent overfitting, but this advantage cannot be reflected in the val set. This finding suggests a trade-off where a simple data balancing on training set cannot simultaneously improve the model's spatial understanding and achieve high evaluation scores.

In contrast, the SQA3D dataset is already balanced. As a result, we observe that over half of the questions correspond to answers that appear more than 790 times, and applying our data balancing method does not lead to performance gains.

Nevertheless, as presented in the right part of Table 5, results on the SQA3D dataset remain largely insensitive to the model architecture. This indicates the model is still leveraging textual shortcuts between questions and answers, an observation that aligns with our analysis in Section 5.3 and Section 5.2.

## H.2 Ablation study on model with data balancing

| Balancing | ScanQA | | |
|---|---|---|---|
| max count | B-4 ↑ | C ↑ | R ↑ |
| *No balancing* | | | |
| 790 | 13.67 | 81.43 | 38.37 |
| *Full model* | | | |
| 100 | 13.75 | 83.29 | 37.89 |
| 50 | 14.67 | 84.23 | 38.42 |
| 10 | 14.27 | 84.63 | 40.11 |
| 1 | 9.68 | 66.56 | 30.66 |
| *Only LLM* | | | |
| 100 | 13.31 | 83.32 | 37.51 |
| 50 | 16.32 | 87.94 | 39.32 |
| 10 | 14.18 | 83.23 | 39.72 |
| 1 | 8.91 | 60.85 | 29.03 |

Table 26: **Further model ablation study with data balancing on ScanQA. Left:** Leveraging Qwen2-7B as the LLM backbone, only LLM setup achieve the best performance with 50 max count. **Right:** visualization of ScanQA distribution.

As shown in the left part of Table 26, further model ablation studies on ScanQA reveals that the optimal performance gain is achieved when the data balancing threshold is set to 50. However, at this setting, the model's performance closely approximates that of a pure text-based model, indicating a persistent overfitting to the data.

Conversely, the largest performance disparity between the model with and without multi-modal inputs is observed when the threshold is 1. Despite this increased reliance on the encoder, the model's overall performance is severely degraded at this threshold.

Furthermore, the right part of Table 26 visualize the answer and question distribution of ScanQA, which reveals that over 75% of the answers appear only once, yet these answers collectively account for only approximately 25% of the entire dataset, indicating a significant long-tail distribution. Additionally, it can be observed that the val set exhibits an answer and question distribution that is nearly identical to that of the training set.

## H.3 Additional comparisons to 3D scene-centric methods

As shown in Table 27, after data balancing of simple under-sampling, our model with only LLM achieve state-of-the-art performance on ScanQA BLUE-4 and CIDEr. Notably, our method achieves the highest performance among all baselines while utilizing the less amount of training data. More specially, after data balancing, the model achieve improvements of 1.00, 2.80, and 0.05 on BLUE-4, CIDEr, and ROGUE, respectively. However, when using the only LLM setup, the model can even further enhance its performance on CIDEr, increasing from 84.23 to 89.17.

If we simply perform data balancing of under-sampling, we can only balance the relationship between questions and answers in the text, but it does not affect the relationship between 3D scene inputs and text content. Under the premise that the model may only fit the text content, mere data balancing actually encourages the model to continue modeling only the relationship between questions and answers. This result further calls for attention to whether models can truly achieve scene understanding, rather than merely comparing metrics.

| Method | Balancing | Training Data | ScanQA | | |
|--------|-----------|---------------|--------|--------|--------|
| | | | BLUE-4↑ | CIDEr↑ | ROUGE↑ |
| *Baselines* | | | | | |
| Chat-3D v2 | ✗ | 204k | 7.3 | 77.1 | 40.1 |
| 3D-LLM | ✗ | 675k | 12.0 | 69.4 | 35.7 |
| SceneLLM | ✗ | 690k | 11.7 | 80.0 | 35.9 |
| LEO* | ✗ | 1034k+145k | - | 80.20 | 40.24 |
| LL3DA | ✗ | 145k | 13.53 | 76.69 | 37.31 |
| Grounded 3D-LLM | ✗ | 187k+146k | 13.4 | 72.7 | - |
| LSceneLLM | ✗ | 145k | - | 88.24 | **40.82** |
| *Ours* | | | | | |
| Full model | ✗ | 145k | 13.67 | 81.43 | 38.37 |
| Full model | ✓ | 145k | 14.67 | 84.23 | 38.42 |
| Only LLM | ✓ | 145k | **16.32** | **89.17** | 39.97 |

Table 27: **Performance comparisons to state-of-the-art 3D scene-centric methods.** Adopting Qwen2-7B as LLM backbone, only LLM setup with data balancing achieve the state-of-the-art performance on without any multi-modal inputs. * denotes do not identify question-related objects for the model.

# I    DISCUSSION

## I.1    PERFORMANCE ANALYSIS OF 3D SCENE-CENTRIC METHODS

| Method | ScanRefer(val) | Multi3DRef(val) | Scan2Cap(val) | | | ScanQA(val) | | | SQA3D(test) | |
|--------|----------------|-----------------|---------------|--------|--------|-------------|--------|--------|-------------|--------|
| | mIoU | mIoU | B-4@0.5 | C@0.5 | R@0.5 | B-4 | C | R | EM | EM-R |
| *Full dataset* | | | | | | | | | | |
| Official 3D-LLaVA (7B) | 43.3 | 42.7 | 36.9 | 78.8 | 57.7 | 17.1 | 92.6 | 43.1 | 54.5 | 56.6 |
| Reproduce | 0 | 8.2 | 28.9 | 30.3 | 52.9 | 11.2 | 61.4 | 32.0 | 45.0 | 46.9 |
| No point cloud feat | 0 | 8.2 | 28.0 | 28.1 | 52.1 | 12.7 | 63.9 | 32.6 | 47.8 | 49.9 |
| No superpoint | 0 | 8.2 | 27.8 | 15.9 | 51.8 | 13.7 | 78.9 | 38.2 | 50.8 | 53.3 |
| No 3D tokens | 0 | 8.2 | 26.9 | 30.3 | 51.7 | 13.7 | 77.9 | 37.9 | 51.5 | 53.9 |
| *(Scan2Cap, ScanQA, SQA3D)* | | | | | | | | | | |
| Reproduce | - | - | 26.8 | 27.0 | 51.7 | 11.3 | 64.6 | 32.8 | 46.6 | 48.8 |
| No point cloud feat | - | - | 27.1 | 23.2 | 51.8 | 11.4 | 67.7 | 34.6 | 48.8 | 50.8 |
| No superpoint | - | - | 27.8 | 15.9 | 51.8 | 14.5 | 81.4 | 39.0 | 52.0 | 54.0 |
| No 3D tokens | - | - | 26.1 | 30.4 | 51.1 | 13.8 | 80.7 | 38.8 | 52.5 | 54.6 |

Table 28: **Further ablation of 3D scene-centric methods.** Take 3D-LLaVA as the representative, which results shows low utilization of its meticulously designed encoder.

Our analysis of 3D scene-centric VLMs aligns with the findings of this paper. First, 3D-LLM (Hong et al., 2023) performance is notably weak compared with other 3D scene-centric approaches, even falling below that of models without a 3D encoder in this paper, likely due to differences in training methodologies. Second, Grounded 3D-LLM (Chen et al., 2024c), despite significant effort in training an object-alignment scene encoder, shows limited performance gains on ScanQA, consistent with our observations in  Section 3.2. Finally, LSceneLLM (Zhi et al., 2024) achieves improved ScanQA performance through finer-grained feature selection. We attribute this to the text-based attention weights used for identifying 3D tokens, which effectively enriches the text distribution and implicitly enhances 3D scene understanding while considering the text, thus mitigating overfitting to high-frequency answer distributions.

As shown in Table 28, we further conduct ablation study on SOTA 3D scene-centric method, 3D-LLaVA. Unfortunately, we can not reproduce its performance. Therefore, we conduct experiments only on 3D-DC and 3D-QA for completeness. The results further corroborate our findings that 3D-LLaVA exhibits low utilization of its meticulously designed 3D Encoder. Moreover, as we progressively remove the additional information beyond text, we observe a gradual increase in the

model's performance on the ScanQA's CiDEr metric, reaching 80.7, which aligns with our experiment results of 81.43 on Qwen2-7B.

## I.2 PERFORMANCE ANALYSIS OF 3D OBJECT-CENTRIC METHODS

| Method | ScanRefer | | Multi3DRef | | Scan2Cap | | ScanQA | | SQA3D |
|---|---|---|---|---|---|---|---|---|---|
| | Acc@0.25 | Acc@0.5 | F1@0.25 | F1@0.5 | B-4@0.5 | C@0.5 | C | EM | EM |
| ***Full model*** | | | | | | | | | |
| Official ChatScene (7B) | 55.5 | 50.2 | 57.1 | 52.4 | 36.3 | 77.1 | 87.7 | 21.6 | 54.6 |
| Reproduce | 55.1 | 49.8 | 57.4 | 52.9 | 33.3 | 72.6 | 83.4 | 19.8 | 54.2 |
| ***Ablation with 2D feat*** | | | | | | | | | |
| Reproduce - 3D scene feat | 52.1 | 46.1 | 54.8 | 49.6 | 32.6 | 70.9 | 83.8 | 19.6 | 54.1 |
| further - scene locs | 52.6 | 46.2 | 55.1 | 49.7 | 32.5 | 70.1 | 82.1 | 19.5 | 54.2 |
| further - obj ids | 52.6 | 46.2 | 55.1 | 49.7 | 32.4 | 69.9 | 82.1 | 19.4 | 54.2 |
| further - assigned ids | 4.8 | 4.5 | 10.4 | 10.2 | 28.0 | 16.1 | 68.6 | 20.0 | 50.9 |
| ***Ablation without 2D feat*** | | | | | | | | | |
| Reproduce - 3D scene feat | 52.1 | 46.1 | 54.8 | 49.6 | 32.6 | 70.9 | 83.8 | 19.6 | 54.1 |
| further - scene img feat | 4.8 | 4.5 | 8.9 | 8.7 | 28.0 | 16.1 | 67.3 | 15.6 | 50.4 |
| further - scene locs | 4.8 | 4.5 | 8.9 | 8.7 | 28.0 | 16.1 | 67.5 | 15.6 | 50.3 |
| further - obj ids | 4.8 | 4.5 | 8.9 | 8.7 | 28.0 | 16.1 | 67.4 | 15.6 | 50.3 |
| further - assigned ids | 4.8 | 4.5 | 8.9 | 8.7 | 28.0 | 16.1 | 68.3 | 15.4 | 49.8 |

Table 29: **Further ablation of 3D object-centric methods.** Take ChatScene as the representative, which results shows high utilization of 2D image feature and assigned ids instead of 3D object encoder.

The model designs for 3D scene-centric VLM and 3D object-centric VLM share considerable similarities. A key distinction lies in their approach to feature extraction: 3D scene-centric VLM employs a 3D scene encoder to extract global scene features, subsequently deriving local object proposal features. Conversely, 3D object-centric VLM starts by extracting local object features and then aggregates them to obtain global scene understanding. The commonality of the model architecture suggests that 3D object-centric VLMs may encounter similar limitations.

Recent advancements in 3D object-centric VLM (Huang et al., 2023b) have demonstrated impressive performance. However, observations from LSceneLLM (Zhi et al., 2024) indicate a potential bottleneck of them. When the prior knowledge of task-relevant object identities is removed from the recognition model, the performance of LEO (Huang et al., 2023b) drops to a level comparable to LL3DA, despite LEO utilizing an $8\times$ larger dataset. This finding aligns with our findings that these models may lack data scaling capabilities on large scale datasets. Furthermore, it implies that a primary advantage of 3D object-centric VLMs stems from the available semantic information associated within defined objects.

As shown in Table 29, among the input of ChatScene, ChatScene heavily relies on 2D scene image features and assigned IDs. During the ablation process, as long as these two elements are maintained, ablation will not significantly affect the model's performance. Therefore, it can be observed that ChatScene is more dependent on the foundation model's prediction of scene images feature and object IDs, rather than on the 3D object encoder and subsequent relational modeling.

While ChatScene theoretically benefits from a semantically rich 3D object encoder (Zhou et al., 2023; Yu et al., 2022; Zhang et al., 2023a; 2022; Zhu et al., 2023b; Huang et al., 2023c; Zeng et al., 2023; Xue et al., 2023; 2024; Li et al., 2025; Liu et al., 2023b), our ablation studies reveal that it does not leverage this component as effectively as Video-3D-LLM. We attribute this discrepancy to their architectural differences. As a 3D object-centric method, ChatScene first detects objects and encodes their individual semantic information. However, it then requires a dedicated relation module to infer the relationships between these objects. This additional step introduces a potential bottleneck: while the initial 3D object features are semantically rich, there is no guarantee that this richness is preserved after passing through the relation module.

This hypothesis is supported by ChatScene's heavy reliance on 2D scene image features. As we concluded in Section 5.3, models tend to favor information that is more direct and easier to utilize. In this case, if the 2D image features offer richer and more accessible semantic content for the LLM

compared to the processed 3D features, the model will naturally depend on the 2D input for spatial understanding. Consequently, despite its design as a 3D object-centric framework, ChatScene's performance characteristics suggest it functions more like a 2D image-based method.

### I.3 Performance analysis of 2D image-based methods

| Method | ScanRefer | | Multi3DRef | | Scan2Cap | | ScanQA | | SQA3D |
|---|---|---|---|---|---|---|---|---|---|
| | Acc@0.25 | Acc@0.5 | F1@0.25 | F1@0.5 | B-4@0.5 | C@0.5 | C | EM | EM |
| *2D image-based 3D VLM* | | | | | | | | | |
| Official Video-3D-LLM (7B) | 58.1 | 51.7 | 58.0 | 52.7 | 41.3 | 83.8 | 102.1 | 30.1 | 58.6 |
| Reproduce | 55.3 | 47.4 | 56.2 | 49.3 | 39.3 | 80.0 | 100.7 | 29.7 | 58.3 |
| No 2D Encoder | 20.7 | 16.5 | 16.1 | 13.5 | 34.3 | 46.9 | 79.6 | 24.1 | 51.8 |
| Only LLM | 23.3 | 18.5 | 23.7 | 19.6 | 30.1 | 35.1 | 76.0 | 21.7 | 51.4 |

Table 30: **Further ablation of 2D image-based methods.** Take Video-3D-LLM as the representative, which results shows high utilization of 2D image encoder.

As shown in Table 30, Video-3D-LLM demonstrates the most ideal results during the ablation process, meaning that performance significantly declines when the 2D image encoder is removed. Furthermore, it can be observed that even only LLM setup can maintain a certain level of 3D-VG capability.

We attribute the effectiveness of Video-3D-LLM to its powerful pre-trained visual encoder that provides semantically rich features, which can be directly used after projector. This essentially simplifies the task for the LLM, allowing it to focus primarily on learning relationships within these high-level, text-aligned representations. Our experiments corroborate this hypothesis. We observe that when pre-training is insufficient to imbue the features with strong semantic meaning, the model tends to disregard these multi-modal inputs. Conversely, when the inputs can be effectively mapped to the textual latent space, the model demonstrates a significantly improved ability to leverage cross-modal information for a comprehensive understanding.

Therefore, it is plausible that Video-3D-LLM would encounter the inherent limitations of both 3D scene-centric and object-centric methods if it were to use DINO as the visual encoder. This is because DINO, despite being a powerful self-supervised pre-trained model, excels at capturing rich textural information from images but is deficient in the semantic content necessary for 2D image-based VLM.

### I.4 LLM usage

We thank the Gemini 2.5-Flash for assistance in editing and polishing the manuscript, including grammar checks, sentence structure refinement, and improving overall clarity. The use of this tool did not introduce any new scientific content or ideas. The authors take full responsibility for all content and claims presented in this work.

### I.5 Social impact

The development of robust 3D VLMs holds promise for a wide range of beneficial applications. These include enhanced human-computer interaction in AR/VR environments, improved scene understanding for autonomous navigation in robotics and self-driving vehicles, and more effective training tools in embodied AI simulations. However, the technology also presents potential risks for negative societal impacts. For example, the capacity of these models to process and interpret detailed 3D scene information could be misused for surveillance purposes, enabling more sophisticated tracking and monitoring of individuals within private or public spaces. Consequently, our analysis provides a renewed understanding of 3D VLMs only within the academic context.

