# OpenReview forum: "Does Your 3D Encoder Really Work? A simple yet effective pathway to real 3D scene understanding"
_ICLR.cc/2026/Conference — Submitted to ICLR 2026_

### Official Review · Reviewer_egbm · 2025-10-29

**Soundness:** 2
**Presentation:** 2
**Contribution:** 2
**Rating:** 4
**Confidence:** 4

**Summary:**

The paper analyzes why 3D scene-centric vision-language models (VLMs) underperform compared to 3D object-centric and 2D-based models, finding that they rely too much on language cues and neglect 3D scene input. To address this, the authors propose rearranging the input sequence to place the 3D scene between the question and answer, forcing the model to use visual information. This simple strategy improves genuine 3D understanding and model performance.

**Strengths:**

1. The authors report several interesting findings on why 3D scene-centric models underperforms to 3D object-centric and 2D-based models. These findings are worth exploration.

2. The authors conduct comprehensive experiments on studying the phenomenons.

3. The proposed sequence rearrangement is simple and seems to be effective in Table 11.

**Weaknesses:**

1. Recent relevant works, such as VG-LLM [1] and 3DRS [2], are not cited or compared in the paper. Including these comparisons would help contextualize the contributions.

2. The paper’s presentation is unsatisfactory. While several subsections are used to introduce findings, there is little explanation of why these phenomena occur or how the proposed method addresses them. Sections 3 and 4, in particular, are tedious and detract from the reader’s engagement and interest.

3. The effectiveness of the proposed sequence rearrangement is not fully validated. The authors should conduct experiments to specifically verify whether this approach addresses the data scaling issue. If such experiments are already included, please clarify or direct me to the relevant section.

4. The authors attribute the observed phenomena to the order of scene, question, and answer tokens. However, in the Video-3D-LLM training set, I found that the token order is also [scene, question, answer]. What accounts for the difference in results—why does this arrangement work for Video-3D-LLM?

[1] Learning from Videos for 3D World: Enhancing MLLMs with 3D Vision Geometry Priors. NeurIPS 2025.

[2] MLLMs Need 3D-Aware Representation Supervision for Scene Understanding. NeurIPS 2025.

**Questions:**

1. Please cite and discuss the relevant recent papers.

2. Please provide more detailed explanations for the observed phenomena.

3. Please report the results of the missing experiments.

4. Please clarify why the [scene, question, answer] token order is effective in Video-3D-LLM.

---

> ### Author Response · Authors · 2025-11-19
>
> Thank you for your careful and insightful comments. We are glad to hear that you appreciate our comprehensive experiments regarding 3D scene-centric models and find our proposed sequence rearrangement simple and effective. In the following, we will address your concerns in detail:
>
> >
> >* __W2,Q2(Further clarification)__ : "...  there is little explanation of why these phenomena occur or how the proposed method addresses them ..."
> >
>
> Thanks for the comment and we next briefly restate the process of discovering, verifying, and solving problems in the paper.
>
> 1. Three key observations:
>
> (1) Instability and Data Scaling Issues (Sec. 3.2): Our initial experiments revealed that the 3D scene-centric approach exhibited instability and limited data scaling capabilities. This was particularly noticeable when leveraging pre-training datasets for multi-modal alignment.
>
> (2) Minimal Pre-training Contribution (Sec. 3.3): We further discovered that the pre-training stage offered no significant contribution to the final performance of the model.
>
> (3) Ineffective 3D Encoder Utilization (Sec. 3.4): Crucially, our analysis indicates that the 3D scene-centric method did not effectively utilize the 3D scene encoder. We found that similar performance could be achieved even in an Only LLM setting.
>
> `Key issue`: Multi-modal LLMs ignore multi-modal 3D inputs but only model pure-text distribution.
>
> 2. Further verification:
>
> (1) Consistent Underutilization Across Benchmarks (Sec. 4.1): We extended our analysis to additional benchmarks, VSI-bench and SQA3D, and confirmed the same fundamental issue: the model exhibits minimal utilization of the 3D Scene Encoder. This is evident as the model's performance remains comparable to an Only LLM setting, effectively suggesting that it primarily relies on the LLM to model the pure-text Question-Answer relationship rather than grounding its responses in the 3D features.
>
> (2) Paradigm-Specific Utilization Discrepancy (Sec. 4.2): We further validated methods adhering to other paradigms. Our results indicate that 3D object-centric methods also face the same problem of ineffective encoder utilization. In contrast, 2D image-based methods demonstrated high and efficient utilization of their respective visual encoders. This highlights a specific challenge within methods that rely on 3D scene features.
>
> (3) Alignment Potential vs. SFT Transfer Failure (Sec. F and G in appendix): Finally, we verified the preliminary alignment capability of the 3D scene Encoder. The pre-training stage successfully establishes a basic ability to map 3D scene features into the text latent space. Crucially, however, this acquired capability did not effectively transfer to the downstream Supervised Fine-Tuning stage. We also confirmed through targeted tests that data format issues were not the cause of this performance bottleneck.
>
> `Key phenomena`: Multi-modal LLMs learns linguistic shortcuts purely from the text distribution, frequently over-predicting answers that appear with high frequency within the training dataset.
>
> 3. Problem solution:
>
> (1) Limited Impact of Data Balancing (Sec. 5.2): We initially attempted data balancing on the dataset. While this approach resulted in marginal performance improvements, it failed to address the core problem of the model overlooking the 3D features.
>
> (2) Solution via Sequence Rearrangement (Sec. 5.3): Our effective solution involves intentionally disrupting the linguistic shortcut present in the standard [$Scene,Question,Answer$] training sequence (specifically the [$Question,Answer$] linguistic correlation). By implementing a rearrangement strategy, we successfully forced the model to re-focus on the 3D inputs, resulting in the desired effective utilization of the scene encoder.
>
> `Key problem`: The core problem lies in the semantic poverty of current 3D inputs, which the LLM cannot directly process. While the pre-training stage successfully converts preliminary cross-modal inputs into the text latent space, the 3D scene training paradigm (one scenario corresponds to multiple QA questions) and the imbalance of the dataset introduce shortcuts that cause the model to ignore the valuable cross-modal features. This leads to the issues observed above.
>
> `Key solution`: Breaking the pure text subset [$Question,Answer$] through the sequence rearrangement to force the model to refocus on alignment and utilize multimodal information.

---

> ### Author Response · Authors · 2025-11-19
>
> >
> >* __W1,Q1(Comparisons to recent SOTA)__ : "Recent relevant works, such as VG-LLM and 3DRS, are not cited or compared in the paper. ..."
> >
>
> | LLM | ScanQA C | ScanQA EM | SQA3D EM | Scan2Cap C@0.5 | Scan2Cap B-4@0.5 | ScanRefer Acc@0.25 | ScanRefer Acc@0.5 | Multi3DRefer F1@0.25 | Multi3DRefer F1@0.5 |
> | --- | --- | --- | --- | --- | --- | --- | --- | --- | --- |
> | Video 3D LLM | 100.7 | 29.7 | 58.3 | 80.0 | 39.3 | 55.3 | 47.4 | 56.2 | 49.3 |
> | w/ DINOv2 | 81.84 | 23.65 | 52.06 | 49.79 | 34.08 | 52.80 | 46.76 | 52.32 | 47.59 |
> |  |  |  |  |  |  |  |  |  |  |
> | 3D-RS | 101.61 | —— | —— | 84.78 | 41.99 | 62.0 | 55.17 | 60.43 | 55.10 |
> | w/ DINOv2 | 82.75 | 24.25 | 54.36 | 54.47 | 36.77 | 59.5 | 54.53 | 56.55 | 53.39 |
>
> **Table 1**: Further comparisons on 2D image-based approach with DINOv2.
>
> Thank you for the suggestion. We have supplemented citation in the mian paper and further experiments including 3D-RS and VGLLM. As discussed above, we further analyze the performance of 2D image-based approach with strong image encoder without semantic information, DINOv2, for completeness.
>
> After replacing SigLIP with DINOv2, Video-3D-LLM and 3D-RS showe a significant decrease in performance on ScanQA, SQA3D, and Scan2Cap, while there is no significant impact on 3D Grounding tasks due to the training objective of contrastive learning. The similar performance in Table 1 (ScanQA CIDER, Video-3D-LLM 81.84, 3D-RS 82.75 vs. LL3DA 81.34) further confirms that the lack of semantic information is the core issue in existing approaches with 3D inputs.
>
>
> | dataset | VSI-bench | Obj. Count | Abs. Dist | Obj. Size | Room Size | Rel. Dist | Rel. Dir | Route Plan | Appr. Order |
> | --- | --- | --- | --- | --- | --- | --- | --- | --- | --- |
> | **All Model** |  |  |  |  |  |  |  |  |  |
> | S1 | 51.34 | 68.95 | 37.73 | 58.5 | 61.7 | 48.59 | 43.81 | 30.92 | 60.51 |
> | S1+S2 | 59.91 | 69.8 | 52.52 | 67.08 | 66.66 | 64.36 | 78.37 | 48.45 | 32.03 |
> | **Only VGGT** |  |  |  |  |  |  |  |  |  |
> | S1 | 39.49 | 61.8 | 30.75 | 50.56 | 44.23 | 31.12 | 40.3 | 28.35 | 28.8 |
> | S1+S2 | 43.3 | 62.37 | 34.59 | 50.71 | 46.49 | 41.12 | 49.01 | 38.14 | 23.94 |
> | **Only LLM** |  |  |  |  |  |  |  |  |  |
> | S1 | 36.63 | 61.84 | 27.27 | 51.66 | 32.53 | 31.26 | 45.41 | 25.77 | 17.31 |
> | S1+S2 | 41.54 | 61.53 | 32.2 | 54.73 | 28.05 | 41.97 | 45.72 | 38.65 | 30.25 |
>
> **Table 2**: Further experiments on VGLLM with only VGGT encoder.
>
> As shown in Table 2, we further include experiments for ablation study on VGLLM. Results in Table 2 demonstrates similar issue that VGLLM with only VGGT and only LLM achieve similar performance, which furhter confirm that VGGT VGGT also faces similar issue as 3D encoder.

---

> ### Author Response · Authors · 2025-11-19
>
> >
> >* __W3,Q3(Data scaling with rearrangement)__ : "... should conduct experiments to specifically verify whether this approach addresses the data scaling issue ..."
> >
>
> | Dataset | B4 | C | R |
> | --- | --- | --- | --- |
> | ScanQA | 11.69 | 72.38 | 34.07 |
> | +Nr3D | 12.62(+0.93) | 72.88(+0.5) | 34.97(+0.9) |
> | +ScanRefer | 12.86(+0.24) | 76.19(+3.31) | 35.98(+1.01) |
> | +3DLLM QA | 13.42(+0.56) | 76.5(+0.31) | 36.3(+0.32) |
> | +scan2cap&Multi3DRefer | 13.3(-0.12) | 78.12(+1.62) | 36.51(+0.21) |
> | +SQA3D | 14.53(+1.23) | 78.86(+0.74) | 35.79(-0.72) |
> | +3RScanQA | 12.9(-1.63) | 71.82(-7.04) | 34.29(-1.5) |
>
> **Table 3**: Further data scaling with sequence rearrangement. (*) denotes the performance improvement with new dataset.
>
> Thank you for your careful suggestion. We supplement further experiments on data scaling with sequence rearrangement. As shown in Table 3, LL3DA with sequence rearrangement shows smoother and more stable data scaling capabilities.
>
> Furthermore, we find that when scaling with the 3RScan dataset without normal attribute as input for 3D scene encoder, the perfromance of ScanQA suddenly decrease, which indicates that the model is really utilizing the features extracted by the encoder. Comparisons to the performance of Table 1 in the main paper further inspires us: the previous model only learned pure text, which masked this issue that the absence of the normal attribute in 3RScan dataset will affect the data scaling on ScanNet.
>
>
> | Methods | ScanQA-B4 | ScanQA-C | ScanQA-R | VSI-bench Avg. |
> | :---: | :---: | :---: | :---: | :---: |
> | LL3DA | 14.00 | 75.23 | 36.50 | 33.90 |
> |  w/ rearrangement | 15.07 | 77.82 | 36.22 | **41.00** |
> |  w/ rearrangement and data balance | **16.45** | **79.26** | **36.35** |  |
>
> **Table 4**: Further comparisons to baseline on balanced ScanQA test set and VSI-bench.
>
> | Dataset | B4 | C | R |
> | --- | --- | --- | --- |
> | ScanQA | 16.45 | 79.26 | 36.35 |
> | +Nr3D | 17.27(+0.82) | 82.07(+2.81) | 36.80(+0.45) |
> | +ScanRefer | 17.39(+0.12) | 83.65(+1.58) | 37.71(+0.91) |
> | +3DLLM QA | 17.95(+0.56) | 83.96(+0.31) | 38.03(+0.32) |
> | +scan2cap&Multi3DRefer | 16.17(-1.78) | 84.95(+0.99) | 38.07(+0.04) |
> | +SQA3D | 17.99(+1.82) | 87.13(+2.18) | 39.07(+1.0) |
> | +3RScanQA | 14.40(-3.59) | 74.45(-12.68) | 35.77(-3.3) |
>
> **Table 5**: Further data scaling with sequence rearrangement and data balance on test set. (*) denotes the performance improvement with new dataset.
>
> Furthermore, as shown in Table 24 in Sec. J of the appendix, the original ScanQA test set itself suffers from data imbalance, which unfortunately allows models to achieve inflated performance by learning language shortcuts. To rigorously demonstrate this effect and validate our model's superiority, we performe an additional ablation study on a balanced version of the ScanQA test set while removing the data balancing on train set in Table 4.
>
> As shown in Table 4, the baseline model's performance on the balanced test set significantly drops and falls far behind our proposed model's performance. This highlights that our model's gains are not due to shortcuts but genuine improvements in 3D-QA reasoning. Therefore, we further include data scaling experiments with balanced test set in Table 5 for completeness.

---

> ### Author Response · Authors · 2025-11-19
>
> >
> >* __W4,Q4(Token order in 2D image-based approach)__ : "... What accounts for the difference in results—why does this arrangement work for Video-3D-LLM?"
> >
>
> | Model | VSI-bench | Obj. Count | Abs. Dist | Obj. Size | Room Size | Rel. Dist | Rel. Dir | Route Plan | Appr. Order |
> | --- | --- | --- | --- | --- | --- | --- | --- | --- | --- |
> | Full Model | **45.4** | **70.0** | **43.6** | **62.2** | **68.4** | **46.7** | **45.2** | 32.2 | 13.8 |
> | No Encoder | 41.5 | 65.6 | 35.6 | 54.5 | 35.3 | 39.4 | 44.6 | **39.1** | 21.5 |
> | Only LLM | 42.9 | 65.6 | 38.7 | 53.2 | 60.9 | 40.5 | 44.9 | 28.1 | **21.8** |
>
> **Table 4**: Further comparisons on 2D image-based approach Video-3D-LLM on VSI-bench.
>
>
> | LLM | Avg. | Obj. Count | Abs. Dist | Obj. Size | Room Size | Rel. Dist | Rel. Dir | Route Plan | Appr. Order |
> | --- | --- | --- | --- | --- | --- | --- | --- | --- | --- |
> | VGLLM official (3B) | 46.1 | 66.4 | **36.6** | **55.2** | 56.3 | 40.8 | 43.4 | **30.4** | 39.5 |
> | +rearrangement | **48.15** | **67.0** | 33.72 | 54.05 | **61.21** | **44.22** | **44.13** | 29.89 | **50.97** |
>
> **Table 5**: Further comparisons on 2D image-based approach VGLLM on VSI-bench.
>
> We sincerely appreciate the your insightful and expansive comment regarding the token order similarity between our work and Video-3D-LLM. Our experimental analysis resolves this concern through three key points:
>
> (1) Limitations of Data Balancing in 2D and 3D: As discussed in the relevant work [1], 2D image-based approaches also face the same problem where the Visual (V) input becomes non-essential in 2D Visual Question Answering (VQA) tasks. While [1] addressed this primarily through data balancing, our work further demonstrates that data balancing is insufficient to resolve this core issue, especially when dealing with the complex 3D inputs.
>
> (2) Evidence of Underutilization in 2D Video-3D-LLM: To directly address the Video-3D-LLM comparison, we analyze its performance on the VSI-bench. As shown in Table 4, we evaluated Video-3D-LLM trained with the VLM-3R dataset. By removing the 32-frame image inputs and the Image Encoder, the performance on VSI-bench only dropped marginally from 45.4% to 42.9%. This result strongly indicates that methods based on 2D images (like Video-3D-LLM) also encountered the same utilization issue, although they ultimately achieved great performance, the visual encoder is not fully utilized.
>
> (3) Generality of the Rearrangement Solution: Furthermore, we test the effectiveness of our proposed rearrangement strategy on VGLLM. As illustrated in Table 5, integrating our rearrangement approach with VGLLM led to a notable 2.05% improvement in accuracy. This not only indicates that VGLLM may also suffer from the same shortcut learning issue but also proves that our rearrangement solution is effective and generalizable to 2D image-based methods.
>
> [1] Elevating the role of image understanding in visual question answering

---

> ### Author Response · Authors · 2025-11-24
>
> Thank you once again for taking the time to review our paper and for providing valuable comments to enhance its quality. We appreciate your thoughtful comments, which have guided us in improving our work. Specifically, we have cited and included comparisons with VGLLM and 3DRS, conducted data scaling experiments with the proposed rearrangement strategy, and provided an analysis of token order in 2D image-based approaches. Your insightful suggestions have been invaluable in helping us address these aspects comprehensively.
>
> We sincerely hope that the additional experiments and our response have addressed your concerns. If you have any further questions or suggestions, we would be glad to offer further clarifications. Thank you for your time and consideration!

---

> ### Comment · Reviewer_egbm · 2025-11-24
>
> Thanks for the rebuttal. However I found the following points are unclear and confusing.
>
> > W1,Q1
>
> In Tab.1, It is confusing that why the authors conduct experiments by replacing SigLIP with DINOv2 on Video-3D-LLM and 3DRS instead of directly comparing with the original model, and how this relates to the study in the main paper.
>
> > W3,Q3
>
> In Tab. 3, it is not clear that how many training samples are used for data scaling experiments. Besides, Tab. 4 misses the result on VSI-Bench of the `w/ rearrangement and data balance` row.

---

> ### Author Response · Authors · 2025-11-24
> **Response to Reviewer egbm (Part 1/2)**
>
> We sincerely thank the reviewer for the prompt feedback and for highlighting areas that required further clarification. We appreciate the opportunity to elaborate on the motivation behind the DINOv2 experimental design and to provide the missing details regarding data scaling and benchmark results. We address these points below:
>
> >
> >* __W1,Q1(Rationale for encoder replacement)__
> >
>
> To facilitate a clear understanding, we first provide the direct answers, followed by a detailed analysis.
>
> **Question1**: Why conduct experiments replacing SigLIP with DINOv2, and how does this relate to the main paper?
>
> **Direct answer1**: To verify that semantic alignment is the critical factor preventing the model from ignoring encoder features.
>
> **Question 2**: Why not directly compare with the original model?
>
> **Direct answer 2**: Strictly comparing with the original model introduces a confounding variable (strong 2D priors of encoder), whereas our goal is to isolate the impact of semantic alignment and explore why the native 3D encoders fail.
>
> ---
>
> In line with our previous discussion (**W2, Q2**), we identified the `Key Problem` as the **semantic poverty** of current 3D inputs. While the original models (equipped with SigLIP/CLIP) achieve high performance, they rely heavily on massive 2D-text alignment, which current native 3D encoders inherently lack. A direct comparison would mask the specific bottlenecks we aim to address: the semantic sparsity of 3D inputs and the limitations of 3D encoders stemming from 3D scene training paradigms and dataset imbalance.
>
> To validate this hypothesis, we designed the comparative experiment in Table 6 to isolate the variable of semantic information:
>
> 1. With Semantic Information: Methods utilizing semantically aligned encoders achieve superior performance.
>
> 2. Without Semantic Information: We replaced the original encoders (SigLIP) with DINOv2 which lacks text-aligned semantics or removed 2D features entirely.
>
> | Methods | ScanQA C | ScanQA EM | SQA3D EM | Scan2Cap C@0.5 | Scan2Cap B-4@0.5 |
> | --- | --- | --- | --- | --- | --- |
> | **With semantic information** |  |  |  |  |  |  |  |  |  |
> | Video-3D-LLM | 100.7 | 29.7 | 58.3 | 80.0 | 39.3 |
> | 3D-RS | 101.61 | —— | —— | 84.78 | 41.99 |
> | ChatScene | 87.7 | 21.6 | 54.6 | 77.1 | 36.3 |
> | **Without semantic information** |  |  |  |  |  |  |  |  |  |
> | Video-3D-LLM w/ DINOv2 | 81.84 | 23.65 | 52.06 | 49.79 | 34.08 |
> | 3D-RS w/ DINOv2 | 82.75 | 24.25 | 54.36 | 54.47 | 36.77 |
> | ChatScene w/o 2D img feat | 80.42 | 18.92 | 52.81 | 61.76 | 31.35 |
> | LL3DA | 81.34 | 21.26 | 51.10 | --- | --- |
>
> **Table 6**: Comparison of isolating the impact of semantic information regardless of training dataset.
>
> **Observation & Conclusion**: As shown in Table 6, when the encoder lacks semantic alignment in the **Without Semantic Info** group, the performance of various state-of-the-art methods drops significantly and converges to a similar baseline (e.g., ScanQA CIDEr ≈ 80, SQA3D EM ≈ 50). Similar trends were observed in our VG-LLM experiments that SigLIP is a more critical factor than VGGT for the final performance.
>
> **This explicitly validate our claim**: The key advantage of recent 2D image-based approaches lies in their semantically aligned encoders, which prevent the LLM from ignoring the input. This justifies the necessity of the proposed method in the main paper to address this semantic gap of 3D encoder.

---

> ### Author Response · Authors · 2025-11-24
> **Response to Reviewer egbm (Part 2/2)**
>
> >
> >* __W3,Q3(Further experiments details)__
> >
>
> | Dataset | B4 | C | R | |
> | --- | --- | --- | --- | --- |
> | ScanQA | 9.47 | 65.56 | 33.89 | 30k |
> | +3D LLM Alignment | 11.69 | 72.38 | 34.07 | 64k |
> | +Nr3D | 12.62(+0.93) | 72.88(+0.5) | 34.97(+0.9) | 105k |
> | +ScanRefer | 12.86(+0.24) | 76.19(+3.31) | 35.98(+1.01) | 145k |
> | +3DLLM QA | 13.42(+0.56) | 76.5(+0.31) | 36.3(+0.32) | 162k |
> | +scan2cap&Multi3DRefer | 13.3(-0.12) | 78.12(+1.62) | 36.51(+0.21) | 263k |
> | +SQA3D | 14.53(+1.23) | 78.86(+0.74) | 35.79(-0.72) | 355k |
> | +3RScanQA & 3RScan Alignment | 12.9(-1.63) | 71.82(-7.04) | 34.29(-1.5) | 661k |
>
> **Table 7**: Further data scaling with sequence rearrangement. (*) denotes the performance improvement with new dataset.
>
>
> | Methods | ScanQA-B4 | ScanQA-C | ScanQA-R | VSI-bench Avg. |
> | :---: | :---: | :---: | :---: | :---: |
> | LL3DA | 14.00 | 75.23 | 36.50 | 33.90 |
> |  w/ rearrangement | 15.07 | 77.82 | 36.22 | **41.00** |
> |  w/ rearrangement and data balance | **16.45** | **79.26** | **36.35** | 39.96 |
>
> **Table 8**: Further comparisons to baseline on balanced ScanQA test set and VSI-bench.
>
>
> We thank the reviewer for the valuable feedback. As shown in Table 7 and Table 8, we have provided additional experimental details. Regarding the results on VSI-bench with our proposed rearrangement, we observe that the performance remains comparable with and without data balancing. We believe this indicates that VSI-bench possesses a higher degree of inherent balance compared to ScanQA.
>
> To clarify our balancing strategy mentioned in table 8, VSI-bench comprises two data formats:
>
> 1. Multiple Choice Questions: These require selecting an answer from options A, B, C, or D. Since this type constitutes the majority of the data, we set an under-sampling threshold of 8,000.
>
> 2. Numerical Questions: These require a specific numerical value as the answer. An inherent advantage of this format is that the model treats proximate values (e.g., 2.1 vs. 2.0) as distinct answers. For this category, we set an under-sampling threshold of 150.
>
> Following this balancing process, the total dataset was reduced to approximately 80% of its original volume. To maintain consistency in training iterations, we re-sampled this balanced subset to match the original dataset size.
>
> We sincerely hope this clarification and the additional results address your concerns. If you have any further questions or suggestions, we would be glad to offer further clarifications. Thank you for your time and consideration!

---

> ### Comment · Reviewer_egbm · 2025-11-25
>
> Thanks for the response.
>
> > In line with our previous discussion (W2, Q2), we identified the `Key Problem` as the semantic poverty of current 3D inputs. While the original models (equipped with SigLIP/CLIP) achieve high performance, they rely heavily on massive 2D-text alignment, which current native 3D encoders inherently lack.
>
> If the `key problem` is the semantic poverty of current 3D inputs, a straightforward way to solve this issue might be enriching the semantic representations along with the 3D encoders, e.g., model ensemble. Have the authors conducted experiments using this naive method and compared its performance with the proposed rearrangement method?

---

> ### Author Response · Authors · 2025-11-25
>
> We thank the reviewer for the highly constructive suggestion. We strongly resonate with the reviewer’s insight regarding the semantic poverty of 3D inputs. To facilitate a clear understanding, we first provide a direct answer, followed by a crucial clarification on experimental fairness, and finally a detailed quantitative analysis.
>
> **Question**: Can model ensemble alleviate the semantic poverty of 3D encoders?
>
> **Direct answer**: Yes, model ensemble can alleviate the performance drop with careful model design to some extent. However, it does not fundamentally solve the issue. Instead, it introduces a new problem: the model tends to take a shortcut by relying almost exclusively on the injected 2D semantics, largely bypassing the 3D geometric information. This defeats the purpose of building a 3D-native understanding.
>
> **Crucial Clarification**: Why we avoid a direct "Apples-to-Apples" comparison. Before discussing the performance results, it is important to clarify why a direct comparison between our rearrangement method and a naive model ensemble will be inherently unfair:
>
> 1. Conflicting Sources of Improvement: Strictly comparing with the ensemble model introduces a massive confounding variable: the strong pre-trained 2D priors. The ensemble method essentially borrows the capabilities of large-scale image-text pre-training, whereas our goal is to isolate and fix the structural alignment of the native 3D encoder.
>
> 2. Goal of our Research: We aim to explore why native 3D encoders fail and how to enable them to learn semantics independently.
>
> ---
>
> Detailed Analysis: To validate the above claim that ensemble models rely on 2D priors rather than learning 3D semantics, we analyzed representative ensemble methods like ChatScene (3D scene feat + 2D image feat) and VGLLM (2D image encoder SigLIP + 3D geometry encoder VGGT). We calculated a simple Feature Contribution Rate $\tau$  to quantify which modality the model actually uses:
>
> $\tau_{A}=\frac{1}{N}\sum^N_{n=1}\frac{Perfromance_{full model}^n - Perfromance_{w/o. feature A}^n}{Perfromance_{full model}^n - Perfromance_{only LLM}^n}$
> ​
>
> where $Perfromance_{full model}^n$ denotes the performance on task $n$ with full model setting and a higher $\tau$ indicates that the model relies more heavily on that feature according to the performance gap.
>
> | Model (Ensemble Strategy)| $\tau_{2D}$ (2D Contribution) | $\tau_{3D}$ (3D Contribution) |
> | --- | --- | --- |
> | ChatScene | 97.97% | 5.26% |
> | VGLLM | 80.55% | 13.86% |
>
> **Table 9**: Ablation study of the feature contribution rate with 2D and 3D feature.
>
> **Interpretation of Results**: This outcome is reasonable and aligns with our previous discussion regarding the model's preference for semantic-rich information. We observe a consistent behavior pattern:
>
> 1. Without Ensemble: When inputs consist only of semantically poor 3D features and text, the model tends to rely excessively on textual priors to answer questions.
>
> 2. With Ensemble: When we inject semantic-rich 2D features via ensemble, the model shifts its focus to these new added information as shortcuts, effectively bypassing the 3D geometry. In both scenarios, the 3D encoder fails to learn robust representations.
>
> ---
>
> | LLM | Avg. | Obj. Count | Abs. Dist | Obj. Size | Room Size | Rel. Dist | Rel. Dir | Route Plan | Appr. Order |
> | --- | --- | --- | --- | --- | --- | --- | --- | --- | --- |
> | VGLLM official (3B) | 46.1 | 66.4 | **36.6** | **55.2** | 56.3 | 40.8 | 43.4 | **30.4** | 39.5 |
> | +rearrangement | **48.15** | **67.0** | 33.72 | 54.05 | **61.21** | **44.22** | **44.13** | 29.89 | **50.97** |
>
> **Table 5**: Further comparisons on 2D image-based approach VGLLM on VSI-bench.
>
> ---
>
> **Conclusion: Complementary and Mutually Reinforcing**
>
> While model ensemble effectively alleviates the performance drop, we maintain that it is a solution that treats the symptoms but not the root cause. It relies on external 2D priors to mask the deficiencies of the 3D encoder, rather than fixing the encoder itself.
>
> However, this does not mean the two approaches are contradictory. On the contrary, they are complementary and synergistic. Since our method fundamentally enhances the semantic alignment of the 3D encoder, it can provide additional gains even within an ensemble framework in Table 5.
>
> **Key Takeaway**: The fact that our method yields further improvements on top of VGLLM demonstrates that we are unlocking native 3D semantic information that the naive ensemble ignored. By fixing the 3D semantic poverty, we allow the model to utilize both the strong 2D priors and the now-aligned 3D features, leading to a mutually reinforcing result.

---

> > ### Comment · Reviewer_egbm · 2025-11-26
> >
> > Thanks for the response.
> >
> > > Instead, it introduces a new problem: the model tends to take a shortcut by relying almost exclusively on the injected 2D semantics, largely bypassing the 3D geometric information. This defeats the purpose of building a 3D-native understanding.
> >
> > Although the authors have compared the contributions of 2D and 3D components, I am still not convinced by their conclusion regarding their respective importance. According to the experimental results in Table 8 of VG-LLM, incorporating 3D geometric information can effectively improve the performance of 2D models. From my perspective, 3D features act as auxiliary cues that assist spatial understanding, but they are not the primary factors, which may explain their relatively smaller contribution to overall performance.
> >
> > In Table 5, the authors used VGLLM-3B for experiments, but only the 4B model is presented in the original VG-LLM paper. Furthermore, the reported baseline performance differs from that of VG-LLM-4B in the original paper. Therefore, I am not convinced by the results shown in Table 5.
> >
> > Thanks.

---

> ### Author Response · Authors · 2025-11-26
>
> We sincerely thank the reviewer for the continued engagement and the in-depth discussion regarding our work. We appreciate the opportunity to clarify the details of our experimental settings and to further discuss the roles of 2D and 3D components in our model. We apologize for any confusion caused by the previous table details and have provided additional clarifications and experimental results below.
>
> >
> >* __(Clarification of table 5)__
> >
>
> | LLM | Avg. | Obj. Count | Abs. Dist | Obj. Size | Room Size | Rel. Dist | Rel. Dir | Route Plan | Appr. Order |
> | --- | --- | --- | --- | --- | --- | --- | --- | --- | --- |
> | Official VGLLM-4B (Qwen2.5-vl 3B + VGGT-1B) | 47.3 | 66.0 | **37.8** | **55.2** | 59.2 | **44.6** | **45.6** | **33.5** | 36.4 |
> | Reproduce | 46.1 | 66.4 | 36.6 | **55.2** | 56.3 | 40.8 | 43.4 | 30.4 | 39.5 |
> | Reproduce w/ rearrangement | **48.15** | **67.0** | 33.72 | 54.05 | **61.21** | 44.22 | 44.13 | 29.89 | **50.97** |
>
> **Table 5**: Further comparisons on 2D image-based approach VGLLM on VSI-bench.
>
> We apologize for the ambiguity in our previous notation. In our previous response, the label "VGLLM-3B" referred specifically to the LLM backbone used (Qwen2.5-VL-3B). When combined with the visual encoder (VGGT-1B), the total parameter count corresponds to the 4B model described in the original VG-LLM paper. There is no discrepancy in the model architecture, and it was a naming issue in our draft.
>
> To address the concern regarding baseline performance, we have updated Table 5. We now explicitly compare our reproduced results with the official results reported in the original VG-LLM paper. As shown, our method consistently achieves performance gains over the official baselines, demonstrating the effectiveness of our approach.
>
> >
> >* __(Discussion on the role of 3D Encoders)__
> >
>
> | dataset | VSI-bench | Obj. Count | Abs. Dist | Obj. Size | Room Size | Rel. Dist | Rel. Dir | Route Plan | Appr. Order |
> | --- | --- | --- | --- | --- | --- | --- | --- | --- | --- |
> | Official VGLLM-8B (Qwen2.5-vl 7B + VGGT-1B) | 62.2 | 71.4 | 56.8 | 69.0 | 69.1 | 67.9 | 83.2 | 47.4 | 32.5 |
> | Reproduce | 59.91 | 69.8 | 52.52 | 67.08 | 66.66 | 64.36 | 78.37 | 48.45 | 32.03 |
> | PI3[1] | 59.59 | 69.96 | 54.59 | 68.27 | 69.61 | 62.53 | 81.65 | 47.42 | 22.65 |
>
> **Table 10**: Further ablation experiments on 3D geometry encoder of VGLLM-8B (Qwen2.5-vl-7B + VGGT-1B) with S1+S2.
>
> We thank the reviewer for this insightful comment. We agree that in many current multimodal systems, 3D features often act merely as auxiliary cues due to the dominance of pre-trained 2D semantics. However, we respectfully argue that this observation precisely highlights the motivation of our work: to bridge the semantic gap in 3D representations to enable true 3D-native understanding.
>
> If we do not inject semantics into the 3D encoder and rely solely on geometric features, the integration with LLMs becomes unpredictable and suboptimal. To demonstrate this, we conducted an ablation study shown in Table 10, where we replaced VGGT with Pi3, a better 3D geometry encoder known for superior geometric representation capabilities compared to VGGT.
>
> Surprisingly, despite Pi3 possessing stronger geometric encoding capabilities, the overall performance was slightly inferior to the VGGT-based baseline in our task. This result supports our hypothesis:
>
> 1. Geometry alone is insufficient: Simply having a stronger geometric encoder, like Pi3, does not guarantee better multimodal performance if the semantic alignment is missing.
>
> 2. The necessity of semantic injection: The "shortcut" via 2D semantics is currently necessary because raw 3D encoders lack the semantic richness to "talk" to the LLM directly. Our method attempts to fix this by enriching the 3D representation.
>
> Therefore, while we acknowledge that 2D features currently play a dominant role, our approach represents a necessary step towards spatial intelligence. By improving how 3D encoders handle semantics, we aim to transition 3D information from an "auxiliary cue" to a primary, native modality for the LLM.
>
> [1] Pi3: Permutation-Equivariant Visual Geometry Learning
>
> **Conclusion**
>
> We are deeply grateful for the reviewer's constructive criticism, which has pushed us to refine our arguments and experimental validation. We hope these clarifications regarding the model architecture in Table 5 and the ablation analysis of 3D encoders in Table 10 satisfactorily address your concerns. Our goal remains to advance the field towards models that can genuinely leverage 3D spatial information.
>
> Sincerely,
> The Authors

---

### Official Review · Reviewer_4oGj · 2025-10-30

**Soundness:** 2
**Presentation:** 2
**Contribution:** 3
**Rating:** 6
**Confidence:** 3

**Summary:**

This paper carries out a study that analysis the usefulness of the 3D encoder in 3D VLMs. The analysis is performed in the aspects of the scaling capabilities of 3D VLMs, the effectiveness of the pretraining stage, and the ablation on the 3D VLMs completely without the 3D encoder. The analysis shows that the 3D encoders in the current scene-centric 3D VLMs are not working properly, and the model might overfit to the pure text question-answer data. To solve this issue, the paper proposes to rearrange the sequence of 3D features and text tokens during training, to force the 3D VLMs to focus on the information from the 3D encoder. Experimental results demonstrate that sequence rearrangement can help with the 3D understanding capability of 3D VLMs.

**Strengths:**

- The paper presents a pretty interesting finding that 3D tokens in current 3D VLMs are not functioning properly, resulting in the 3D VLMs even achieving similar results without the presence of 3D features. This is an important finding and can raise the awareness of this issue for the community.

- The remedy of rearranging the input sequence is a simple but effective solution to solve the spotted issue, ensuring the 3D encoder to properly function in 3D reasoning.

**Weaknesses:**

- As the paper acknowledges on this point, there is performance gap of the model studied in the paper and the current state-of-the-arts. I think this could be a valid concern because the current state-of-the-art models may already have mitigated issues in these points. If this is the case, the significance of the issue studied in the paper will be downplayed.

- This paper follows a problem analysis followed by proposed method workflow, which is often great for understanding the paper. However, I feel the motivation of the analysis part is not very natural. For example, it is unclear why the study will start from inspecting whether 3D VLMs have scaling capabilities, as it seems to have loose relationship with the issue spotted in the paper.

- Typo: The title of Section 3.3 "Dose" should be "Does".

**Questions:**

- Is the studied issue in the paper still being a severe issue in more recent 3D VLMs? Basically for the 3D VLMs that the authors refer as state-of-the-arts which has large performance gap with the more primitive models studied in the paper.

- The issue studied in the paper is most severe for 3D scene-centric 3D VLMs, with mitigated effects on 3D object-centric 3D VLMs and 2D image-based 3D VLMs. The experiment with the proposed rearranging strategy is also carried out in 3D scene-centric 3D VLMs. How is the performance comparison between the 3D scene-centric VLMs after applying the rearranging strategy and the other types of 3D VLMs?

---

> ### Author Response · Authors · 2025-11-19
>
> Thank you for your thoughtful comments and positive recognition of our important finding regarding 3D token malfunction. We are also glad that you appreciate our proposed approaches as a simple but effective solution to ensure proper 3D reasoning. We will address your concerns point by point:
>
> >
> >* __W1(Effectiveness of sequence rearrangement)__ : "... there is performance gap of the model studied in the paper and the current state-of-the-arts ..."
> >
>
> Thank you for the comment. We conduct further experiments to address your concerns in detail.
>
> 1. _Comparisons to 3D scene-centric baseline_
>
> | Methods | ScanQA-B4 | ScanQA-C | ScanQA-R | VSI-bench Avg. |
> | :---: | :---: | :---: | :---: | :---: |
> | LL3DA | 14.00 | 75.23 | **36.50** | 33.90 |
> | LL3DA w/ rearrangement | **15.07** | **77.82** | 36.22 | **41.00** |
>
> **Table 1**: Further comparisons to baseline on balanced ScanQA test set and VSI-bench.
>
> While we kept the ScanQA test set unchanged in the main paper for a consistent comparison, as shown in Table 24 in Sec. J of the appendix, the original ScanQA test set itself suffers from data imbalance, which unfortunately allows models to achieve inflated performance by learning language shortcuts. To rigorously demonstrate this effect and validate our model's superiority, we performe an additional ablation study on a balanced version of the ScanQA test set while removing the data balancing on train set.
>
> As shown in Table 1, the baseline model's performance on the balanced test set significantly drops and falls far behind our proposed model's performance. This highlights that our model's gains are not due to shortcuts but genuine improvements in 3D-QA reasoning. Furthermore, we have included a supplementary comparison on a more balanced and challenging dataset, VSI-bench. After training with our proposed rearrangement, our model shows a significant performance boost. In contrast, the original baseline model's performance remains stagnant, barely exceeding the VSI-bench Chance level of 34.0.
>
> Also as shown in Sec. I.1 in the appendix, our experiments reveals that the recent 3D scene-centric approach, 3D-LLaVA is still affected by this issue.
>
> 2. _Ablation study on state-of-the-art models_
>
> | LLM | ScanQA C | ScanQA EM | SQA3D EM | Scan2Cap C@0.5 | Scan2Cap B-4@0.5 | ScanRefer Acc@0.25 | ScanRefer Acc@0.5 | Multi3DRefer F1@0.25 | Multi3DRefer F1@0.5 |
> | --- | --- | --- | --- | --- | --- | --- | --- | --- | --- |
> | Video 3D LLM | 100.7 | 29.7 | 58.3 | 80.0 | 39.3 | 55.3 | 47.4 | 56.2 | 49.3 |
> | w/ DINOv2 | 81.84 | 23.65 | 52.06 | 49.79 | 34.08 | 52.80 | 46.76 | 52.32 | 47.59 |
> |  |  |  |  |  |  |  |  |  |  |
> | 3D-RS | 101.61 | —— | —— | 84.78 | 41.99 | 62.0 | 55.17 | 60.43 | 55.10 |
> | w/ DINOv2 | 82.75 | 24.25 | 54.36 | 54.47 | 36.77 | 59.5 | 54.53 | 56.55 | 53.39 |
>
> **Table 2**: Further comparisons on 2D image-based approaches with DINOv2.
>
>
> With successful pre-trained model like SigLIP, 2D image-based approachs are able to map image into text latent space. As shown in Table 2, we replace the SigLIP with strong image encoder without image-text alignment, DINOv2. The results in ScanQA, SQA3D, and Scan2Cap show a significant decrease, while 3D Growth was not significantly affected by the training paradigm. Furthermore, we can observe that the performance of 2D image-based approaches with DINOv2 in ScanQA CIDER is closed to that of LL3DA (Video-3D-LLM's 81.84 vs. LL3DA's 81.34). Therefore, without aligned semantic information, state-of-the-art models will suffer from the same issue Unfortunately, to our best knowledge, there is currently no works that can provide stable and effective semantic information for 3D inputs.
>
> | Model | VSI-bench | Obj. Count | Abs. Dist | Obj. Size | Room Size | Rel. Dist | Rel. Dir | Route Plan | Appr. Order |
> | --- | --- | --- | --- | --- | --- | --- | --- | --- | --- |
> | Full Model | **45.4** | **70.0** | **43.6** | **62.2** | **68.4** | **46.7** | **45.2** | 32.2 | 13.8 |
> | No Encoder | 41.5 | 65.6 | 35.6 | 54.5 | 35.3 | 39.4 | 44.6 | **39.1** | 21.5 |
> | Only LLM | 42.9 | 65.6 | 38.7 | 53.2 | 60.9 | 40.5 | 44.9 | 28.1 | **21.8** |
>
> **Table 3**: Further comparisons on 2D image-based approach Video-3D-LLM on VSI-bench.
>
> As shown in Table 3, we supplyment further experiments on Video-3D-LLM on VSI-bench. Video-3D-LLM trained with data from VLM-3R also occur the same issue that similar performance is shown between Full Model and Only LLM setting.

---

> ### Author Response · Authors · 2025-11-19
>
> >
> >* __W2(Paper clarification)__ : "...  it is unclear why the study will start from inspecting whether 3D VLMs have scaling capabilities ..."
> >
>
> | Full Model | B4 | C | R | Only LLM  | B4 | C | R |
> | --- | --- | --- | --- | --- | --- | --- | --- |
> | 30k | 11.12 | 70.06 | 36.47 |  | 10.68 | 73.12 | 37.74 |
> | 64k | 10.90  | 70.88  | 36.00 |  | 9.14 | 73.31 | 37.09 |
> | 105k | 12.46 | 73.49  | 36.82 |  | 9.40 | 73.91 | 37.39 |
> | 145k | 13.33 | 77.23  | 37.10 |  | 9.67 | 74.74 | 37.88 |
> | 162k | 13.63 | 77.38  | 36.80 |  | 10.33 | 77.20 | 38.01 |
> | 263k | 12.87 | 76.42 | 36.56 |  | 11.14 | 76.84 | 38.46 |
> | 355k | 13.64 | 78.56 | 37.44 |  | 11.51 | 74.98 | 37.00 |
> | 661k | 12.65 | 77.31 | 37.43 |  | 9.62 | 70.63 | 36.73 |
>
> **Table 4**: Comparisons between Full Model and Only LLM setting on scaling capabilities.
>
> Thank you for your thoughtful comments. We have further supplemented experiments of Only LLM (right part) setting to clarify our motivation on scaling capabilities. In this part we want to introduce three key question:
>
> 1. Inconsistent scaling and intuitive contradiction: 3D scene-centric methods fail to achieve stable and effective performance gains through data scaling. Intuitively, knowledge acquired from one task (e.g., understanding object descriptions via 3D Dense Captioning) should aid performance in another (3D-QA). The observed lack of cross-task scaling contradicts this intuition and challenges the scaling laws.
>
> 2. Performance gap: The failure to boost performance via data expansion raised the key question: Why do 3D scene-centric methods significantly lag behind 2D image-based methods? If data scaling is ineffective, the issue must lie deeper within the model's utilization of the input.
>
> 3. Similar performance: As the scale of leveraged dataset increases, Full Model setting and Only LLM setting exhibit similar performance and trends of change, which contradicts our intuition.
>
> We sincerely thank you again for the clarification of the paper and we will refine the this part after rebuttal.
>
>
> >
> >* __W3(Typo refinement)__ : "Typo: The title of Section 3.3 "Dose" should be "Does"."
> >
>
> Thank you for the suggestion. We have now refined this typo in the main paper.

---

> ### Author Response · Authors · 2025-11-19
>
> >
> >* __Q1(Comparisons to recent 3D VLMs)__ : "Is the studied issue in the paper still being a severe issue in more recent 3D VLMs?"
> >
>
>
> | Method | ScanRefer (mIoU) | Multi3DRef (mIoU) | S2C (B-4@0.5) | S2C (C@0.5) | S2C (R@0.5) | SQA (B-4) | SQA (C) | SQA (R) | SQA3D (EM) | SQA3D (EM-R) |
> | :--- | :---: | :---: | :---: | :---: | :---: | :---: | :---: | :---: | :---: | :---: |
> | _**Full dataset**_ | | | | | | | | | | |
> | **Official 3D-LLaVA (7B)** | **43.3** | **42.7** | **36.9** | **78.8** | **57.7** | **17.1** | **92.6** | **43.1** | **54.5** | **56.6** |
> | Reproduce | 0 | 8.2 | 28.9 | 30.3 | 52.9 | 11.2 | 61.4 | 32.0 | 45.0 | 46.9 |
> | No point cloud feat | 0 | 8.2 | 28.0 | 28.1 | 52.1 | 12.7 | 63.9 | 32.6 | 47.8 | 49.9 |
> | No superpoint | 0 | 8.2 | 27.8 | 15.9 | 51.8 | 13.7 | 78.9 | 38.2 | 50.8 | 53.3 |
> | Only LLM | 0 | 8.2 | 26.9 | 30.3 | 51.7 | 13.7 | 77.9 | 37.9 | 51.5 | 53.9 |
> | _**(Scan2Cap, ScanQA, SQA3D)**_ | | | | | | | | | | |
> | Reproduce | - | - | 26.8 | 27.0 | 51.7 | 11.3 | 64.6 | 32.8 | 46.6 | 48.8 |
> | No point cloud feat | - | - | 27.1 | 23.2 | 51.8 | 11.4 | 67.7 | 34.6 | 48.8 | 50.8 |
> | No superpoint | - | - | 27.8 | 15.9 | 51.8 | 14.5 | 81.4 | 39.0 | 52.0 | 54.0 |
> | Only LLM | - | - | 26.1 | 30.4 | 51.1 | 13.8 | 80.7 | 38.8 | 52.5 | 54.6 |
>
> **Table 5**: Further ablation of 3D scene-centric methods 3D-LLaVA, which results show low utilization of its meticulously designed encoder.
>
>
> | dataset | VSI-bench | Obj. Count | Abs. Dist | Obj. Size | Room Size | Rel. Dist | Rel. Dir | Route Plan | Appr. Order |
> | --- | --- | --- | --- | --- | --- | --- | --- | --- | --- |
> | **All Model** |  |  |  |  |  |  |  |  |  |
> | S1 | 51.34 | 68.95 | 37.73 | 58.5 | 61.7 | 48.59 | 43.81 | 30.92 | 60.51 |
> | S1+S2 | 59.91 | 69.8 | 52.52 | 67.08 | 66.66 | 64.36 | 78.37 | 48.45 | 32.03 |
> | **Only VGGT** |  |  |  |  |  |  |  |  |  |
> | S1 | 39.49 | 61.8 | 30.75 | 50.56 | 44.23 | 31.12 | 40.3 | 28.35 | 28.8 |
> | S1+S2 | 43.3 | 62.37 | 34.59 | 50.71 | 46.49 | 41.12 | 49.01 | 38.14 | 23.94 |
> | **Only LLM** |  |  |  |  |  |  |  |  |  |
> | S1 | 36.63 | 61.84 | 27.27 | 51.66 | 32.53 | 31.26 | 45.41 | 25.77 | 17.31 |
> | S1+S2 | 41.54 | 61.53 | 32.2 | 54.73 | 28.05 | 41.97 | 45.72 | 38.65 | 30.25 |
>
> **Table 6**: Further experiments on VGLLM with only VGGT encoder.
>
>
> Thank you for the comments. To our best knowledge, 3D-LLaVA is the latest approach with pure 3D input. Unfortunately, as shown in table 5, we may not reproduce its results and we further report results only with Scan2Cap, ScanQA, SQA3D for completeness. Furthermore, we further supplement experiments on VGLLM with only VGGT for completeness.
>
> Results in Table 5 and Table 6 reveals the same insight, with 3D encoder without semantic information through image-text alignment, performance will significantly decrease and approach the performance of Only LLM setting. More specially, as the components of 3D LLaVA are gradually removed, ScanQA CIDER even gradually increases, ultimately achieving performance similar to other methods (3D-LLaVA's 80.7 vs. Video-3D-LLM's 81.84, LL3DA's 81.34). Resutls in Table 6 demonstrates that, compared to the performance of the original model, VGGT only contributed 5.5% and 2.9% of the performance, respectively
>
> >
> >* __Q2(Comparisons to 3D object-centric approaches)__ : "... How is the performance comparison between the 3D scene-centric VLMs after applying the rearranging strategy and the other types of 3D VLMs?"
> >
>
> |  | ScanQA-B4 | ScanQA-C | ScanQA-R |
> | --- | --- | --- | --- |
> | **With balance**   |  |    |    |
> | Chat-Scene | 10.81 | 80.42 | 39.43 |
> | Ours | **14.67** | **84.23** | **38.42** |
> | **With rearrangement**   |  |    |    |
> | Chat-Scene  | 6.56 | 60.84 | 31.89 |
> | Ours | **9.41** | **66.55** | **34.67** |
> | **With balance and rearrangement**   |  |    |    |
> | Chat-Scene | 8.69 | 67.79 | 32.16 |
> | Ours  | **11.69** | **72.38** | **34.07** |
>
> **Table 7**: Further comparisons to 3D object-centric approach ChatScene on ScanQA.
>
> Thank you for the comments. To further illustrate the effectiveness of our proposed approach, we conduct comparisons to ChatScene in Table 7. For a fair comparison, we remove the 2D image feature of ChatScene to exclude the influence of semantic information.
>
> As shown in Table 7, our approach outperform ChatScene in various setting, which reveal that ChatScene present a focus on 2D image but not the 3D object.

---

> ### Author Response · Authors · 2025-11-24
>
> Thank you once again for taking the time to review our paper and for providing valuable comments to enhance its quality. We appreciate your thoughtful feedback, which has guided us in improving our work. Specifically, we have included additional experiments on current state-of-the-art models, analyzed data scaling capabilities, and added comparisons with 3D object-centric approaches. Your insightful suggestions have been invaluable in helping us address these aspects comprehensively.
>
> We sincerely hope that the additional experiments and our response have addressed your concerns. If you have any further questions or suggestions, we would be glad to offer further clarifications. Thank you for your time and consideration!

---

> > ### Comment · Reviewer_4oGj · 2025-11-24
> >
> > Dear authors,
> >
> > Thank you for providing the response! However, I do not feel my questions are properly addressed, and the authors seem not to answer many of my questions in the straightforward way. Let me clarify that:
> >
> > Essentially W1 & Q1 are the same question. I was asking if the issue already gets mitigated in recent 3D VLMs, which the authors admit in the paper that they do not show the state-of-the-art 3D VLMs (Lines 481-482). So my concern is, if the issue is already largely mitigated in current state-of-the-arts, then the importance of this study will get largely downplayed. However, the authors only showed 3D-LLaVA and LL3DA in the response, which already appeared in the original manuscript. Also, I do not understand why the authors would mention some DINO-related stuff in response to my question, as I feel it is irrelevant to my question.
> >
> > For Q2, as the authors say (Lines 79-100) that current 3D VLMs can be classified into three categories: 3D scene-centric
> > VLM, 3D object-centric VLM, and 2D image-based VLM. Among them, the authors find that 3D scene-centric VLMs have the issue of skipping 3D features, so the study of the paper focuses on this point. So my question is, for the other two types of VLMs (3D object-centric VLM and 2D image-based VLM), how severe this issue is compared to 3D scene-centric VLMs? Will this issue get mitigated, or get magnified? However, the authors show in the response is a single model ChatScene with or without balance and rearrangement, which I also find irrelevant to my question.

---

> ### Author Response · Authors · 2025-11-25
>
> We sincerely thank the reviewer for the follow-up comments. We apologize that our previous response was not straightforward enough and may have caused confusion regarding the relevant comparisons. To facilitate a clear understanding, we first provide the direct answers, followed by a detailed analysis：
>
> >
> >* __Q1(Is this issue already largely mitigated in current state-of-the-arts?)__
> >
>
> **Direct Answer**: No. The issue persists across all dominant paradigms, including 3D scene-centric, 3D object-centric, and the more recent 2D image-based approaches (current SOTAs).
>
> **Detailed Analysis**: To address your concern regarding the current SOTAs, we categorize current SOTAs into two types and analyze them respectively:
>
> 1. Current SOTA with Pure-3D Inputs (3D-LLaVA): To the best of our knowledge, 3D-LLaVA remains the representative state-of-the-art method for pure 3D point cloud inputs. Our proposed method outperforms it (ScanQA CIDEr: 72.38 vs. reproduced 3D-LLaVA 61.4). This substantial margin demonstrates that the issue addressed is indeed critical in pure 3D baselines and validates the effectiveness of our solution. Furthermore, as evidenced in Tables 1 and 7, our model incorporating the rearrangement mechanism also surpasses LL3DA and ChatScene, providing a comprehensive comparison with both 3D scene-centric and 3D object-centric approaches.
>
> | LLM | Avg. | Obj. Count | Abs. Dist | Obj. Size | Room Size | Rel. Dist | Rel. Dir | Route Plan | Appr. Order |
> | --- | --- | --- | --- | --- | --- | --- | --- | --- | --- |
> | VGLLM official (3B) | 46.1 | 66.4 | **36.6** | **55.2** | 56.3 | 40.8 | 43.4 | **30.4** | 39.5 |
> | +rearrangement | **48.15** | **67.0** | 33.72 | 54.05 | **61.21** | **44.22** | **44.13** | 29.89 | **50.97** |
>
> **Table 8**: Further comparisons on 2D image-based approach VGLLM on VSI-bench.
>
> 2. Current SOTA with 2D Image Inputs: We acknowledge a performance gap between our method and recent 2D image-based approaches (e.g., Video-3D-LLM, VGLLM) that utilize 2D images to achieve SOTA performance. These methods benefit significantly from pre-trained 2D encoders with strong linguistic alignment, whereas current 3D scene encoders lack comparable semantic richness. In Table 2, we substituted SigLIP with DINOv2 to isolate the impact of linguistic alignment. The results confirm that the encoder's alignment capability is a key factor. Nevertheless, Tables 3 and 8 demonstrate that the underlying issue persists even with a robust 2D encoder, and our rearrangement mechanism proves essential for boosting the final performance.
>
> **Conclusion for Q1**: The issue is not mitigated in current SOTAs. It is a fundamental challenge caused by 3D scene training paradigm and dataset imbalance that affects both pure 3D and 2D image-based paradigms.
>
> >
> >* __Q2(Comparison of severity among different approaches.)__
> >
>
> **Direct Answer**: Severity: 3D Scene-centric ≈ 3D Object-centric > 2D Image-based. Both 3D Scene-centric and Object-centric approaches suffer severely from this issue, while 2D Image-based approaches are relatively less affected but still significant.
>
> Detailed Analysis:
>
> 1. **3D Scene-centric VLMs (Severe)**: These models rely on 3D scene encoders that lack large-scale linguistic alignment pre-training. Consequently, they are highly susceptible to the shortcut learning issue caused by 3D scene training paradigms and dataset imbalance, as analyzed in our paper.
>
> 2. **3D Object-centric VLMs (Severe)**: Although they utilize strong 3D object encoders with linguistic alignment, a single object encoder cannot represent a holistic scene. These methods must lift individual objects into a scene representation (injecting IDs, positions, and relationships). Crucially, this object-to-scene lifting process typically lacks semantic supervision,  leading to the same severity of feature skipping as scene-centric models.
>
> 3. **2D Image-based VLMs (Less Severe)**: As mentioned above, these models benefit from powerful 2D encoders (e.g., CLIP, SigLIP) that align visual features with text during pre-training. This alignment acts as a buffer, making them less prone to skipping features compared to pure 3D methods, though the issue persists.
>
> **Conclusion for Q2**: We observe a hierarchy of severity where 3D Scene-centric ≈ 3D Object-centric > 2D Image-based. The fundamental challenge lies in the 3D domain's lack of large-scale linguistic alignment for 3D scene encoders or semantic supervision during scene composition for 3D object encoders, unlike 2D methods which benefit from aligned pre-training like CLIP.
>
> We hope this straightforward explanation address your concerns. If you have any further questions or suggestions, we would be glad to offer further clarifications. Thank you for your time and consideration!

---

> > ### Comment · Reviewer_4oGj · 2025-11-26
> >
> > Dear authors,
> >
> > Thank you for making the answers for clear. For Q2, I think having a table in the paper measuring the comparisons of these three types of approaches would be helpful (I think now 3D scene-centric VLMs are already in the paper, and 2D image-based VLMs are presented in Table 8 in the above rebuttal, so we also need 3D object-centric VLMs to finish this table). I think it would be interesting because the paper makes this categorization for 3D VLMs beforehand.

---

> ### Author Response · Authors · 2025-11-27
>
> We sincerely thank you for this constructive suggestion, which has significantly improved the readability and comprehensiveness of our paper. We agree that a unified comparison across these three paradigms provides valuable insight for the readers.
>
> Following your advice, we have refined the manuscript to include new comparison tables (Table 10 and 11 in the revised paper). This table now presents holistic evaluations of 3D scene-centric, 2D image-based, and 3D object-centric VLMs, integrating the results from our previous rebuttal. To accommodate this detailed analysis within the main text and maintain a clear narrative flow, we have moved the experiments regarding data scaling to the Appendix C.
>
> We greatly appreciate your guidance, which has helped us better position our work within the broader literature. We thank you for holding a positive view of our contributions. If you have any additional comments, we would be glad to offer further clarifications.
>
> Thank you!
>
> Sincerely,
> The Authors

---

### Official Review · Reviewer_8U3K · 2025-10-31

**Soundness:** 2
**Presentation:** 3
**Contribution:** 2
**Rating:** 4
**Confidence:** 3

**Summary:**

This paper first analyzes the limitations of the 3D scene-centric VLMs in the 3D scene QA task. The authors find that the model scales poorly with more data, and has limited reliance on the 3D scene information, as the performance is on par with pure LLM taking only the question as input. The authors attribute this to the imbalance of the answer distribution, and propose to solve it by rearranging the input sequence (positioning the 3D scene between the question and the answer).

**Strengths:**

1. The paper studies an important problem of scene-level 3D understanding using VLMs. It systematically analyzes the shortcuts existing in baselines due to the imbalance in the answer distribution, and has an interesting finding that question-only LLMs can do as well as VLMs with scene input.
2. The paper is easy to follow. It is well-written in general despite some minor typos. The experiment protocols and datasets are clearly described and seem reproducible.

**Weaknesses:**

The analysis of the problem makes sense to me, but my main concern of this paper is that it doesn't propose a valid solution, and the contribution is not enough. The data-balancing helps, but that is a well-known technique. The position swapping between the scene and the questions is more interesting, but the results are not promising. See below:
1. One of the main contributions that this paper claims is swapping the positions of 3D scene and question, which the authors claim to help the model "achieving genuine visual understanding". However, the performance of the final model is worse than the model without swapping (Compare Tab. 11 to Tab. 3). It is unclear what does genuine visual understanding actually mean, and why does it matter to us if the model's performance is even worse. One experiment the authors could do is to evaluate the zero-shot or few-shot generalization capability of both models on other benchmarks, where hopefully the baseline will fail due to overfitting, whereas the proposed model could generalize better.
2. It is unclear what is the 3D encoder architecture, and how much does the capacity of the encoder matters to the conclusion the authors draw. It would be helpful to compare the results across different encoder backbones, such as Point Transformer v2 and v3.
3. I'm confused by Tab 7 (left). All trained models are worse than the random guess by frequency baseline, even the full model trained by the authors. What is the takeaway from this table?

**Questions:**

1. In theory, why would the order of the question and the 3D scene tokens matter to the VLM, since the VLM is an attention-based architecture that models all-pairs relationships? Doesn't the shortcut from text to answer still exist?

---

> ### Author Response · Authors · 2025-11-19
>
> Thank you for your detailed and careful feedback on our paper. We are glad to hear that you appreciate our systematic analysis of shortcuts in scene-level 3D understanding and find the paper easy to follow. In the following, we will address your concerns in detail:
>
> >
> >* __W1(Effectiveness of sequence rearrangement)__ : "... the performance of the final model is worse than the model without swapping ..."
> >
>
> Thank you for the comment regarding the effectiveness of our rearrangement strategy and the validity of our proposed solution. We supplement further experiments to address your concerns in detail.
>
> 1. _Comparisons to 3D scene-centric baseline_
>
> | Methods | ScanQA-B4 | ScanQA-C | ScanQA-R | VSI-bench Avg. |
> | :---: | :---: | :---: | :---: | :---: |
> | LL3DA | 14.00 | 75.23 | **36.50** | 33.90 |
> | LL3DA w/ rearrangement | **15.07** | **77.82** | 36.22 | **41.00** |
>
> **Table 1**: Further comparisons to baseline on balanced ScanQA test set and VSI-bench.
>
> While we kept the ScanQA test set unchanged in the main paper for a consistent comparison, as shown in Table 24 in Sec. J of the appendix, the original ScanQA test set itself suffers from data imbalance, which unfortunately allows models to achieve inflated performance by learning language shortcuts. To rigorously demonstrate this effect and validate our model's superiority, we performe an additional ablation study on a balanced version of the ScanQA test set while removing the data balancing on train set.
>
> As shown in Table 1, the baseline model's performance on the balanced test set significantly drops and falls far behind our proposed model's performance. This highlights that our model's gains are not due to shortcuts but genuine improvements in 3D-QA reasoning. Furthermore, we have included a supplementary comparison on a more balanced and challenging dataset, VSI-bench. After training with our proposed rearrangement, our model shows a significant performance boost. In contrast, the original baseline model's performance remains stagnant, barely exceeding the VSI-bench Chance level of 34.0.
>
> 2. _Comparisons to 3D object-centric approach_
>
> |  | ScanQA-B4 | ScanQA-C | ScanQA-R |
> | --- | --- | --- | --- |
> | **With balance**   |  |    |    |
> | Chat-Scene | 10.81 | 80.42 | 39.43 |
> | Ours | **14.67** | **84.23** | **38.42** |
> | **With rearrangement**   |  |    |    |
> | Chat-Scene  | 6.56 | 60.84 | 31.89 |
> | Ours | **9.41** | **66.55** | **34.67** |
> | **With balance and rearrangement**   |  |    |    |
> | Chat-Scene | 8.69 | 67.79 | 32.16 |
> | Ours  | **11.69** | **72.38** | **34.07** |
>
> **Table 2**: Further comparisons to 3D object-centric approach ChatScene on ScanQA.
>
> To further illustrate the effectiveness of our proposed approach, we conduct comparisons to ChatScene in Table 2. For a fair comparison, we remove the 2D image feature of ChatScene.
>
> As shown in Table2, our approach outperform ChatScene in various setting, which reveal that ChatScene present a focus on 2D image but not the 3D object.
>
> 3. _Comparisons on 2D image-based approach_
>
> | LLM | Avg. | Obj. Count | Abs. Dist | Obj. Size | Room Size | Rel. Dist | Rel. Dir | Route Plan | Appr. Order |
> | --- | --- | --- | --- | --- | --- | --- | --- | --- | --- |
> | VGLLM official (3B) | 46.1 | 66.4 | **36.6** | **55.2** | 56.3 | 40.8 | 43.4 | **30.4** | 39.5 |
> | +rearrangement | **48.15** | **67.0** | 33.72 | 54.05 | **61.21** | **44.22** | **44.13** | 29.89 | **50.97** |
>
> **Table 3**: Further comparisons on 2D image-based approach VGLLM on VSI-bench.
>
> The major advantage of the 2D image-based approach lies in the semantic information provided by the pre-trained encoder, which is currently lacking in 3D scene-based and 3D object-based approaches. For a fair comparison, we directly apply rearrangement on 2D image-based approach.
>
> As shown in Table 3, VGLLM achieves a performance improvement of 2.05 on VSI-bench, which further demonstrates the effectivness of rearrangement.

---

> ### Author Response · Authors · 2025-11-19
>
> >
> >* __W3(Details of experiments on VSI-bench)__ : "I'm confused by Tab 7 ... What is the takeaway from this table?"
> >
>
> Thank you for the comment. The main takeaway from Table 7 is precisely the failure of existing methods to utilize the 3D Scene Encoder, which verify our findings.
>
> The results in Table 7 support two key observations:
>
> 1. Ablation Insensitivity: We observe minimal performance degradation across different model settings on VSI-bench. The model's accuracy changes only slightly, from 33.9% for the _Full model_ setting down to 31.8% for the _Only LLM_ setting setting. This insensitivity immediately suggests that the model is failing to leverage the 3D encoder and that performance primarily stems from the _Only LLM_ component's reliance on memorizing question-answer relationship.
>
> 2. Proximity to Chance Level: The overall accuracy and the accuracy across every sub-category are strikingly close to the VSI-bench chance level. This proximity confirms that the models lack true 3D understanding and are prone to making random guesses based on linguistic cues rather than spatial reasoning.
>
> This poor baseline performance forms the foundation for our contribution. As demonstrated in Table 1, models with our proposed rearrangement strategy achieve a significant performance boost on the challenging VSI-bench benchmark. This stark contrast between the baselines in Table 1 effectively validates our approach's ability to successfully engage the 3D encoder and resolve the underlying issue.
>
> Furthermore, we next briefly summarize our contributions. We sincerely appreciate the opportunity to summarize our contributions.
>
> Our work centers on the discovery, validation, and subsequent resolution of a critical issue prevalent in existing spatial intelligence approaches utilizing 3D inputs. Our proposed rearrangement is an simple strategy applicable to any multi-modal model. Its core mechanism is to shuffle the distribution of input tokens, effectively serving as token-level data augmentation designed to force the model to refocus on true multi-modal correlations rather than unimodal shortcuts. The effectiveness of this strategy is strongly supported by our empirical results.
>
>
> >
> >* __Q1(Theory explanation)__ : "... why would the order of the question and the 3D scene tokens matter to the VLM ..."
> >
>
>
>
> Thank you for this insightful and fundamental question. While Attention does offer global receptive fields, several factors lead the model to still favor local dependencies and exploit textual shortcuts.
>
> 1. The Role of Positional Embedding
>
> Although the Attention mechanism allows for all-pairs relationships, components like Positional Embeddings (Pos Emb) inherently bias the model toward using tokens that are closer in the sequence for the next-token prediction task. In the default ordering, the close proximity of Question and Answer tokens naturally strengthens this local dependency.
>
> 2. Influence of LLM Pre-training
>
> LLMs like Qwen are extensively pre-trained on massive text corpora, equipping them with deep knowledge and ample QA experience. This makes the direct Question-to-Answer mapping a more natural and readily available shortcut during fine-tuning, especially when compared to the complex task of cross-modal alignment.
>
> 3. Training Pattern and Data Skew
>
> As discussed in Section 7 of main paper, several real-world factors push the model toward the text-only shortcut within the [$Scene,Question,Answer$] sequence: (a) a single 3D scene corresponds to multiple QA pairs, (b) the 3D encoder lacks effective pre-training alignment, and (c) the dataset suffers from significant imbalance. These factors make learning the [$Question,Answer$] linguistic pattern a much simpler optimization path than integrating the complex [$Scene$] information.
>
> 4. The Persistence of the Question-to-Answer Shortcut
>
> The shortcut fundamentally persists because the objective of deep learning is to minimize loss. When the dataset exhibits significant shortcuts or imbalances, the LLM will robustly learn the path of least resistance—the text-only shortcut—regardless of the Attention mechanism's global view. As shown in Table 3, results reveal that
> there are certain shortcuts in existing 2D image-based approachs.
>
> Our work block the shorcuts in 2. and 3. through simple data balancing and sequence rearrangement. Its core mechanism is to shuffle the distribution of input tokens, effectively serving as token-level data augmentation designed to force the model to refocus on true multi-modal correlations rather than unimodal shortcuts. By breaking the established [$Question,Answer$] proximity, we compel the model to utilize the displaced [$Scene$] token sequence for accurate prediction.
>
> We are optimistic that future work can build upon this insight by focusing on both data and architectural diversity to fully leverage the benefits of the Attention mechanism without falling into local optimum dominated by textual shortcuts.

---

> ### Author Response · Authors · 2025-11-19
>
> >
> >* __W2(Comparisons with different 3D encoder)__ : "... It would be helpful to compare the results across different encoder backbones ..."
> >
>
> | 3D Encoder | B4 | C | R |
> | --- | --- | --- | --- |
> | Vote2Cap-Detr++[1] | **13.67** | 81.34 | 38.37 |
> | PointTransformerV3[2] | 13.45 | 80.68 | 38.13 |
> | Sonata[3] | 13.14 | **84.16** | **38.98** |
>
> **Table 4**: Comparisons on ScanQA with other 3D encoder backbones.
>
> Thank you for your insightful comments for the completeness of our comparisons. As shown in Table 4, our main experiments leverage Vote2Cap-Detr++ as 3D scene encoder following LL3DA and we supplement further comparisons with PointTransformerV3 and Sonata. Results in table 4 demonstrates that leveraing latest 3D point cloud encoder, Sonata, improve performance on CIDER and ROUGE.
>
> However, despite the observed performance increase when utilizing LL3DA with sonata, a significant gap still persists when compared to leading 2D image-based methods. For instance, Video-3D-LLM achieves a substantially higher CIDEr score of 102.1. While switching to a 3D encoder demonstrably contributes to performance improvements, it is clear that this change alone does not resolve the most fundamental issues underlying 3D multi-modal spatial intelligence.
>
>
> [1] Vote2cap-detr++: Decoupling localization and describing for end-to-end 3d dense captioning, 2024, IEEE TPAMI
>
> [2] Point Transformer V3: Simpler, Faster, Stronger, 2024 CVPR
>
> [3] Sonata: Self-Supervised Learning of Reliable Point Representations, 2025 CVPR

---

> ### Author Response · Authors · 2025-11-24
>
> Thank you once again for taking the time to review our paper and for providing valuable comments to enhance its quality. We appreciate your thoughtful feedback, which has guided us in improving our work. Specifically, we have included experiments for more comprehensive comparisons, added further comparisons with PointTransformerV3 and Sonata, clarified the insights from the data scaling experiments, and provided further analysis of pure-text shortcuts. Your insightful suggestions have been invaluable in helping us address these aspects comprehensively.
>
> We sincerely hope that the additional experiments and our response have addressed your concerns. If you have any further questions or suggestions, we would be glad to offer further clarifications. Thank you for your time and consideration!

---

### Official Review · Reviewer_xho9 · 2025-11-01

**Soundness:** 2
**Presentation:** 2
**Contribution:** 1
**Rating:** 4
**Confidence:** 3

**Summary:**

This work investigates the phenomenon of limited reliance on the 3D scene encoder in 3D scene-centric vision-language models. The authors analyze how these models scale with pre-training data and identify that the 3D encoder is often underutilized. To address this, they propose a sequence rearrangement strategy for the input, demonstrating its effectiveness on the ScanQA benchmark.

**Strengths:**

1. The paper provides a thorough investigation into the underutilization of the 3D scene encoder, a timely problem for the field.
2. The extensive experiments on data scaling offer diagnostic insights into 3D VLMs.

**Weaknesses:**

1. The proposed solution, sequence rearrangement, is introduced only near the end of the paper. Its connection to the earlier analysis is not clearly justified, and the experiments on ScanQA alone are insufficient to convincingly validate its effectiveness.
2. The rearrangement strategy itself is relatively simple and does not constitute a strong methodological contribution.

**Questions:**

1. As noted in Weakness 1, could the authors better explain the reasoning behind why sequence rearrangement helps mitigate the observed issue?
2. It is unclear which datasets were used for the scaling experiments, could the authors clarify this?
3. Could the authors elaborate on how the lack of improvement from data scaling supports the hypothesis that the 3D scene encoder is underutilized?

---

> ### Author Response · Authors · 2025-11-19
>
> Thank you for your careful and insightful comments. We are encouraged that you recognize the timeliness of the problem regarding 3D scene encoder underutilization and the value of our extensive experiments on data scaling. In the following, we will address your concerns in detail:
>
> >
> >* __W1,Q1(Effectiveness of sequence rearrangement)__ : "... why sequence rearrangement helps mitigate the observed issue?"
> >
>
>
> Thank you for the comment. We appreciate you raising the attention regarding the effectiveness of rearrangement and the comprehensive comparison.
>
> `Why sequence rearrangement works`
>
> Thank you for this comment on the mechanism behind sequence rearrangement. Our proposed rearrangement strategy is the direct result of our core discovery: that the model neglects the 3D multi-modal input and learns a purely text-based shortcut. Therefore, the simple and intuitive objective of rearrangement is to disrupt this textual shortcut.
>
> As shown in  Figure 4 in Section 5.3 of the main paper, specifically, the LLM learns a token-level probability distribution based on the input sequence. In the default sequence structure, [$Scene,Question,Answer$], the model primarily relies on the [$Question,Answer$] sequence only for the shortcut, often treating the preceding [$Scene$] token sequence as an irrelevant prefix due to the lack of semantic information.
>
> The core idea of rearrangement is to change this probability distribution by altering the sequence order to [$Question,Scene,Answer$]. By rearranging the input tokens, we intentionally break the most easily learnable textual dependency (i.e., the Question-Answer shortcut). This forces the model to integrate the [$Scene$] tokens effectively, as it can no longer rely on partial sequence contents to learn the shortcut.
>
> Therefore, this mechanism shuffles the distribution of input tokens, effectively serving as token-level data augmentation designed to force the model to refocus on true multi-modal correlations rather than unimodal shortcuts.
>
> `Further comparison with rearrangement`
>
> 1. _Comparisons to 3D scene-centric baseline_
>
> | Methods | ScanQA-B4 | ScanQA-C | ScanQA-R | VSI-bench Avg. |
> | :---: | :---: | :---: | :---: | :---: |
> | LL3DA | 14.00 | 75.23 | **36.50** | 33.90 |
> | LL3DA w/ rearrangement | **15.07** | **77.82** | 36.22 | **41.00** |
>
> **Table 1**: Further comparisons to baseline on balanced ScanQA test set and VSI-bench.
>
> While we kept the ScanQA test set unchanged in the main paper for a consistent comparison, as shown in Table 24 in Sec. J of the appendix, the original ScanQA test set itself suffers from data imbalance, which unfortunately allows models to achieve inflated performance by learning language shortcuts. To rigorously demonstrate this effect and validate our model's superiority, we performe an additional ablation study on a balanced version of the ScanQA test set while removing the data balancing on train set.
>
> As shown in Table 1, the baseline model's performance on the balanced test set significantly drops and falls far behind our proposed model's performance. This highlights that our model's gains are not due to shortcuts but genuine improvements in 3D-QA reasoning. Furthermore, we have included a supplementary comparison on a more balanced and challenging dataset, VSI-bench. After training with our proposed rearrangement, our model shows a significant performance boost. In contrast, the original baseline model's performance is closed to the VSI-bench Chance level of 34.0.
>
> 2. _Comparisons to 3D object-centric approach_
>
>
> |  | ScanQA-B4 | ScanQA-C | ScanQA-R |
> | --- | --- | --- | --- |
> | **With balance**   |  |    |    |
> | Chat-Scene | 10.81 | 80.42 | 39.43 |
> | Ours | **14.67** | **84.23** | **38.42** |
> | **With rearrangement**   |  |    |    |
> | Chat-Scene  | 6.56 | 60.84 | 31.89 |
> | Ours | **9.41** | **66.55** | **34.67** |
> | **With balance and rearrangement**   |  |    |    |
> | Chat-Scene | 8.69 | 67.79 | 32.16 |
> | Ours  | **11.69** | **72.38** | **34.07** |
>
> **Table 2**: Further comparisons to 3D object-centric approach ChatScene on ScanQA.
>
> To further illustrate the effectiveness of our proposed approach, we conduct comparisons to ChatScene in Table 2. For a fair comparison, we remove the 2D image feature of ChatScene to exclude the influence of semantic information.
>
> As shown in Table2, our approach outperform ChatScene in various setting, which reveal that ChatScene present a focus on 2D image but not the 3D object.
>
>
> (Due to space limitations, we continue our response in the following comment.)

---

> ### Author Response · Authors · 2025-11-19
>
> 3. _Comparisons on 2D image-based approach_
>
> | LLM | Avg. | Obj. Count | Abs. Dist | Obj. Size | Room Size | Rel. Dist | Rel. Dir | Route Plan | Appr. Order |
> | --- | --- | --- | --- | --- | --- | --- | --- | --- | --- |
> | VGLLM official (3B) | 46.1 | 66.4 | **36.6** | **55.2** | 56.3 | 40.8 | 43.4 | **30.4** | 39.5 |
> | +rearrangement | **48.15** | **67.0** | 33.72 | 54.05 | **61.21** | **44.22** | **44.13** | 29.89 | **50.97** |
>
> **Table 3**: Further comparisons on 2D image-based approach VGLLM on VSI-bench.
>
>
> As discussed above, the major advantage of the 2D image-based approach lies in the aligned semantic information provided by the pre-trained encoder, which is currently lacking in 3D scene-based and 3D object-based approaches. For a fair comparison, we directly apply rearrangement on 2D image-based approach.
>
> As shown in Table 3, VGLLM achieves a performance improvement of 2.05 on VSI-bench, which further demonstrates the effectivness of rearrangement.
>
>
> >
> >* __Q2(Details of data scaling)__ : "... which datasets were used for the scaling experiments ..."
> >
>
> | LLM | Dataset | BLUE-4 ↑ | CIDEr ↑ | ROUGE ↑ | Details |
> | :---: | :---: | :---: | :---: | :---: | :--- |
> | **Qwen2-1.5B** | 30k | 11.12 | 70.06 | 36.47 | ScanQA |
> |  |  64k  | 10.90 | 70.88 | 36.00 | further +3DLLM Alignment |
> |  | 105k | 12.46 | 73.49 | 36.82 | further +Nr3D |
> |  | 145k | 13.33 | 77.23 | 37.10 | further +ScanRefer (Same setting with LL3DA) |
> |  | 162k | 13.63 | 77.38 | 36.80 | further +3D-LLM QA |
> |  | 263k | 12.87 | 76.42 | 36.56 | further +Multi3DRefer & Scan2Cap |
> |  | 355k | 13.64 | 78.56 | 37.44 | further +SQA3D & 3RScanQA |
> |  | 661k | 12.65 | 77.31 | 37.43 | further +LEO Alignment |
> |---|---|---|---|---|---|
> | **Qwen2-7B** | 30k | 10.61 | 78.69 | 38.93 | ScanQA |
> |  | 145k | 13.67 | 81.34 | 38.37 | same setting with LL3DA |
> |  | 661k | 14.43 | 81.52 | 38.57 | lager scale dataset |
>
> **Table 4**: Details of used dataset on data scaling.
>
> As shown in Table4, we supplement details of used dataset on our data scaling experiments, which indicates that the performance of LL3DA remains almost unchanged across datasets(3D-QA: ScanQA, SQA3D), tasks(3D-QA, 3D-DC), and scenarios(Scannet, 3RScan).
>
>
> >
> >* __Q3(Details of experiments)__ : "... how the lack of improvement from data scaling supports the hypothesis that the 3D scene encoder is underutilized?"
> >
>
>
> Thank you for this insightful question regarding the implications of the data scaling results. Your question highlights a critical challenge we observed, which ultimately led to our core finding. The limited improvement from data scaling initially indicated two major concerns:
>
> 1. Inconsistent scaling and intuitive contradiction: 3D scene-centric methods fail to achieve stable and effective performance gains through data scaling. Intuitively, knowledge acquired from one task (e.g., understanding object descriptions via 3D Dense Captioning) should aid performance in another (3D-QA). The observed lack of cross-task scaling contradicts this intuition and challenges the scaling laws.
>
> 2. Performance gap: The failure to boost performance via data expansion raised the key question: Why do 3D scene-centric methods significantly lag behind 2D image-based methods? If data scaling is ineffective, the issue must lie deeper within the model's utilization of the input.
>
> Our subsequent core finding the underutilization of the 3D scene encoder directly and comprehensively explain these questions. When the 3D Scene Encoder is not effectively engaged, the LLM is learning purely language-based shortcuts derived from the dataset's text distribution, rather than multi-modal cues. This explains the scaling failure:
>
> 1. Scaling hindrance: Scaling is blocked because the LLM is learning a text distribution. Even though 3D-DC and 3D-QA share the same object, their distinct Question-Answer formats prevent effective scaling across tasks. Similarly, different 3D-QA datasets, due to varied Question-Answer distributions, also fail to yield cross-dataset benefits.
>
> 2. The Root Cause of the Gap: The massive performance deficit between 3D scene-centric and 2D image-based methods stems precisely from the 3D models' inability to effectively leverage the rich multi-modal information encoded by the 3D encoder.

---

> ### Author Response · Authors · 2025-11-19
>
> >
> >* __W2(Methodological contribution)__ : "The rearrangement strategy itself is relatively simple and does not constitute a strong methodological contribution."
> >
>
>
> Thank you for the comment. We will briefly summarize our contributions and then discuss them in detail in the next comment. We sincerely appreciate the opportunity to summarize our contributions.
>
> Our work centers on the discovery, validation, and subsequent resolution of a critical issue prevalent in existing spatial intelligence approaches utilizing 3D inputs. Our proposed rearrangement is an simple strategy applicable to any multi-modal model. Its core mechanism is to shuffle the distribution of input tokens, effectively serving as token-level data augmentation designed to force the model to refocus on true multi-modal correlations rather than unimodal shortcuts. The effectiveness of this strategy is strongly supported by our empirical results, as detailed in Tables 1 through 3.

---

> ### Author Response · Authors · 2025-11-19
>
> Thank you once again for providing us with the opportunity to discuss the depth of our methodological contribution. We want to elaborate on the intrinsic motivation and the significant value of this work. We argue that the contribution lies not in the complexity of the operation, but in the critical problem discovery and the minimal, yet highly effective, solution required to fix a foundational flaw.
>
>
> >
> >* __Why 3D Representation is Paramount in Spatial Intelligence？__
> >
>
> We first establish the critical importance of 3D representation. The main limitation of using 2D images for spatial encoding is the loss of real-world scale. A photograph of a real scene can be identical to one of a miniature model, yet their true scales differ drastically. Furthermore, as scenes grow or change, the token count required to represent them effectively using 2D image patches increases prohibitively.
>
> In our view, 3D representation is vital because it intrinsically provides stable spatial structure, real-world scale, computational efficiency, and high potential for future interactivity.
>
>
> >
> >* __The Observed Failure of 3D-Centric Approaches__
> >
>
> Despite 3D's potential, current spatial intelligence research is dominated by 2D image-based methods. While some methods incorporate 3D, their core architectures often rely heavily on 2D VQA frameworks. The lack of high-performing, purely 3D-input methods motivated our investigation: What exactly is preventing 3D from becoming the superior representation for multi-modal LLMs? And what is required to effectively utilize 3D information?
>
>
> >
> >* __Why Existing 3D Geometry Integration Falls Short__
> >
>
> Recent advancements like VLM3R show the promise of integrating 3D geometry (e.g., using VGGT, CUT3R), yet we observed that the performance uplift from geometry encoders is often marginal. For instance, VLM3R's[1] gain on VSI-bench (from 57.74 to 60.90, only 5% performance) suggests that the improvement is not fully unlocking the 3D potential.
>
> [1] VLM-3R: Vision-Language Models Augmented with Instruction-Aligned 3D Reconstruction
>
> >
> >* __Our Core Finding: The Key Problem Supported by Evidence__
> >
>
> Our deep investigation into 3D scene-centric models (like LL3DA) revealed the core problem: as shown in our paper, the model learns a purely text-based shortcut rather than genuine 3D understanding. We were even able to achieve SOTA-level BLUE-4 and CIDEr on ScanQA simply by using data balancing, highlighting the flaw.
>
> The ultimate answer is that LL3DA does not effectively utilize the 3D Scene Encoder. Appendix Section F confirms the encoder possesses alignment capability. Model finds it simpler and more efficient to rely on the pure text QA shortcut than to utilize the cross-modal 3D scene encoder. Consequently, even the pre-training stage fails to engage the 3D information.
>
>
> >
> >* __Our First Attempt to Resolve the Bottleneck__
> >
>
> Recognizing that a long sequence like [$Scene,Question,Answer$] encourages the model to leverage the simpler [$Question,Answer$] subset, our intuitive solution was to disrupt this shortcut.
>
> We propose rearranging the input sequence to [$Question,Scene,Answer$]. The efficacy lies in the fact that while the model might have seen the [$Question,Answer$] pattern before, it is unlikely to have seen the [$Scene,Answer$] pattern. This forces the model to perform cross-modal alignment, mapping the 3D Scene information into the text space for task completion.
>
> The results in Figure 1 underscore this success: Without the 3D input, our model's performance is near zero. Furthermore, we observe significant gains from the pre-training stage—which is designed to better align the scene tokens—confirming that the scene input is now genuinely impacting the final result, with better-aligned models yielding better performance.
>
> In Table 1, we additionally demonstrate that the high performance of the LL3DA baseline on the original ScanQA test set is due to data imbalance and over-memorization of high-frequency terms. Under a fairer comparison on the balanced ScanQA test set (supplementary result) and the more equitable VSI-bench, the effectiveness of our simple rearrangement method is clearly established.
>
> >
> >* __A Call for Rethinking 3D Representation__
> >
>
> We conclude by using this work to call for a re-evaluation of 3D representation in multi-modal LLMs. 3D is effective, but the field currently lacks robust pre-training alignment. Issues include the 3D encoder lacking intrinsic semantics, or subsequent relational modules disrupting the semantics of object encoders. We are optimistic that the next generation of 3D input, equipped with semantic richness, will unlock the full potential of LLM integration.

---

> ### Author Response · Authors · 2025-11-24
>
> Thank you once again for taking the time to review our paper and for providing valuable comments to enhance its quality. We appreciate your constructive feedback, which prompted us to demonstrate our comprehensive performance comparisons and the insights revealed by our data scaling experiments. We are grateful for your thoughtful suggestions, which helped us explore the full potential of our work.
>
> We sincerely hope that the additional experiments and our response have addressed your concerns. If you have any further questions or suggestions, we would be glad to offer further clarifications. Thank you for your time and consideration!

---

### Author Response · Authors · 2025-11-19
**General Response – Thanks to All Reviewers for Constructive and Insightful Feedback**

We greatly appreciate the insightful and constructive feedback provided by all reviewers. We are particularly encouraged by the consensus recognizing the timeliness and significance of the problem we address—the underutilization of 3D scene encoders in current Vision-Language Models (VLMs)—and the diagnostic value of our systematic analysis.

Reviewers xho9, 8U3K, and egbm acknowledged that our paper provides a thorough investigation into an important and timely problem for the 3D VLM community: the failure of models to properly leverage 3D scene features. Reviewer 8U3K found our systematic analysis of linguistic shortcuts to be valuable, noting the intriguing finding that a question-only LLM can achieve performance comparable to 3D scene-input VLMs. Reviewer xho9 emphasized the quality of our extensive experiments, which offer diagnostic insights into the issue.

Reviewers 4oGj and egbm specifically praised our solution. Reviewer 4oGj highlighted that the discovery of 3D tokens "not functioning properly" is an important finding and appreciated that our proposed remedy of rearranging the input sequence is a simple but effective solution for ensuring the 3D encoder functions as intended. Reviewer egbm also recognized that the proposed sequence rearrangement is simple and effective.

We believe that we have been able to thoroughly address all reviewers’ comments by clarifying certain sections of the paper, providing additional experimental verifications, and strengthening the discussion around the novelty and generality of our proposed solution. Details on these changes and specific responses to individual concerns can be found in the reply to each reviewer.

---

### Author Response · Authors · 2025-12-01
**Summary Reply to AC and all Reviewers**

We have received the notification from the ICLR Program Chairs regarding the recent anonymity breach on OpenReview. We hold the integrity of the peer-review process in high regard and commend the organizers for their diligence in maintaining fairness during this incident. We confirm that we have strictly upheld all confidentiality guidelines and have not accessed any non-public information. Although the discussion phase has concluded, we take this opportunity to formally recap the key arguments from our previous responses, ensuring the technical merits of our work are clearly presented.

# Summary of Key Responses to Reviewer Concerns

1. **Comprehensive Benchmark Comparisons.** (Reviewers xho9, 8U3K, & egbm)

- We evaluated our method on balanced datasets (VSI-bench and balanced ScanQA) to demonstrate the effectiveness of the proposed rearrangement strategy.

- We supplemented data scaling experiments with the proposed rearrangement, demonstrating smooth performance improvements as data volume increases.

2. **More Baseline Comparisons** (Reviewers 8U3K, 4oGj, & egbm)

- 3D Encoder: We conducted further comparisons with recent 3D scene encoders (PointTransformerV3 and Sonata) for completeness.

- 3D Scene-centric Approaches: As detailed in Sec. I of the supplementary material, we verified that the latest 3D scene-centric approaches are also significantly impacted by this issue.

- 3D Object-centric Approaches: Under a fair setting of pure 3D understanding, methods using our rearrangement achieve superior performance compared to 3D object-centric approaches.

- 2D Image-based Approaches: We verified that the advantage of these methods largely stems from their strong 2D encoders compared to 3D-based approaches. For a fair comparison, we supplemented VGLLM with our rearrangement to achieve improved performance.

3. **Further Clarification & Paper Refinement** (Reviewers xho9, 8U3K, 4oGj, & egbm)

- We clarified the datasets leveraged for data scaling, validating the assumption that the 3D scene encoder is currently underutilized.

- We provided a detailed analysis of experiments on VSI-bench to verify that methods with pure-3D inputs often fail to achieve true 3D-native understanding.

- Based on the reviewers' constructive comments, we have further optimized the manuscript.

4. **Supplemented Discussion** (Reviewers xho9, 4oGj, & egbm)

- Why does rearrangement alleviate this issue?: The core function of rearrangement is to disrupt text shortcuts, forcing the model to perform pre-trained text alignment, thereby enabling the model to truly leverage the 3D scene.

- Severity of the issue: We provided ample experimental results demonstrating that this problem affects all training paradigms. The severity of the issue follows the order: 3D Scene-centric ≈ 3D Object-centric > 2D Image-based.

- Why does [Scene, Question, Answer] work in 2D image-based approaches?: In line with our previous insights and supplemented experiments, 2D image-based approaches are relatively less affected due to their successful 2D encoders, though the impact remains significant.

# Summary of Discussions

Our work investigates a critical failure in current 3D-native understanding: why existing 3D encoders fail to provide meaningful visual signals. Through extensive experiments, we reveal that despite accepting 3D inputs, models often bypass visual information, relying instead on textual shortcuts. We demonstrate that while LLMs inherently possess the capacity for native 3D perception, prevalent data imbalances and training paradigms drive them toward these text-only dependencies. Crucially, this issue is often masked by high performance on benchmarks like ScanQA, which fail to reflect the lack of true visual grounding. To address this, we propose a simple yet effective rearrangement strategy. This method disrupts textual shortcuts, forcing the model to acquire genuine 3D perception capabilities. Finally, we show that this issue is pervasive across recent state-of-the-art works, highlighting both the severity of the problem and the generality of our solution.

To conclude, we have endeavored to address all major concerns raised by the reviewers—including dataset clarification, manuscript refinement, detailed analysis, and further comparisons—through extensive experiments and clarifications. All rebuttal content, including additional tables, citations, and expanded discussion, will be integrated into the final revision.

We sincerely thank all reviewers and the ACs for their time, insightful feedback, and dedication to scientific rigor. We particularly appreciate the in-depth discussions during the rebuttal phase. While perspectives on specific details may vary, we value the consensus reached regarding the severity and prevalence of the critical issue addressed in our work. Despite the unfortunate platform issues, we remain committed to transparent and constructive scholarly communication.

---

### Meta-Review · Area_Chair_SPex · 2026-01-08

**Summary:**

This work reveals the problem of current 3D VLMs that current models overly rely on language instructions and 3D tokens are often overlooked. To address this problem, the authors propose to rearrange the input sequence by positioning the 3D scene tokens between the question and answer tokens. It receives initial scores of 3 bordreline rejects and 1 borderline accept. Reviewers raised concerns about the technical contribution and experimental validation of the rearrangement approach, the significance of the problem for current models, relationship between problem analysis and proposed approach, and lack of clarity in presentation.

**Reviewer Concerns:**

Some questions related to experimental validation are addressed by more experimental results. While other concerns such as the significance of the problem, relationship between problem analysis and proposed approach, are not fully addressed.

**Reviewer Scores:**

Reviewer 4oGj initially gave the borderline accept score and he/she is convinced in the discussion.

Reviewer egbm initially gave the score of borderline reject. After profound discussions with the authors, he/she is not fully convinced by the authors.

The other two reviewers, initially rated as borderline reject and reject, respectively, are likely to maintain their scores due to some unresolved issues.

---

### Decision · Program_Chairs · 2026-01-26

Reject